# Growth responses of trees and understory plants to nitrogen fertilization in a subtropical forest in China

Di Tian[1], Peng Li[1], Wenjing Fang[1], Jun Xu[2], Yongkai Luo[3], Zhengbing Yan[1], Biao Zhu[1], Jingjing Wang[2], Xiaoniu Xu[2], Jingyun Fang[1]*

[1]*Department of Ecology, College of Urban and Environmental Sciences, and Key Laboratory for Earth Surface Processes of the Ministry of Education, Peking University, Beijing, 100871, China;*

[2]*Department of Forestry, Anhui Agricultural University, 230036, Hefei, Anhui, China;*

[3]*State Key Laboratory of Vegetation and Environmental Change, Institute of Botany, Chinese Academy of Sciences, Beijing 100093, China*

*Correspondence author:

Dr. Jingyun Fang

Department of Ecology, Peking University

Beijing 100871, China

E-mail: jyfang@ urban.pku.edu.cn

**Abstract**

Reactive nitrogen (N) increase in the biosphere has been a noteworthy aspect of global change, producing considerable ecological effects on the functioning and dynamics of the terrestrial ecosystems. A number of observational studies have explored responses of plants to experimentally simulated N enrichment in boreal and temperate forests. Here we asked how the dominant trees and different understory plants respond to experimental N enrichment in a subtropical forest in China. We conducted a 3.4-year N fertilization experiment in an old-aged subtropical evergreen broad-leaved forest in eastern China with three treatment levels applied to nine 20 m×20 m plots and replicated in three blocks. We divided the plants into trees, saplings, shrubs (including tree seedlings), and ground-cover plants (ferns) according to the growth forms, and then measured the absolute and relative basal area increments of trees and saplings and the aboveground biomass of understory shrubs and ferns. We further grouped individuals of the dominant tree species *Castanopsis eyrei* into three size classes to investigate their respective growth responses to the N fertilization. Our results showed that the plot-averaged absolute and relative growth rates of basal area and

aboveground biomass of trees were not affected by N fertilization. Across the individuals of
*C. eyrei*, the small trees with a DBH (diameter at breast height) of 5-10 cm has declined by
66.4% and 59.5%, respectively, in N50 (50 kg N ha$^{-1}$ yr$^{-1}$) and N100 fertilized plots (100 kg
N ha$^{-1}$ yr$^{-1}$), while the growth of median and large trees with a DBH of >10 cm has not
significantly changed with the N fertilization. The growth rate of small trees, saplings and the
aboveground biomass of understory shrubs and ground-cover ferns decreased significantly in
the N fertilized plots. Our findings suggested that N might not be a limiting nutrient in this
mature subtropical forest, and the limitation of other nutrients in the forest ecosystem might
be aggravated by the enhanced N availability, potentially resulting in an adverse effect on the
development of natural subtropical forest.

**Key-words:** *Castanopsis eyrei*, N fertilization, plant growth, shrub layer, subtropical forest,
tree layer, ground-cover fern

## 1 Introduction

Reactive nitrogen (N) increase in the biosphere, especially atmospheric N deposition, is a globally prevalent phenomenon (Galloway *et al*. 2004). It has become a serious environmental issue in China, especially in the southeastern regions, with drastic increase of N entering terrestrial ecosystems which produces considerable ecological effects on the functioning and dynamics of the terrestrial ecosystems (Liu *et al*. 2013; Gu *et al*. 2015). Since the 1990s, the simulated N-fertilization experiments have been conducted in various forest ecosystems to explore the responses of plants and other organisms to the potential high N enrichment and changes of soil N availability (e.g., Wright & Tietema 1995; Bobbink *et al*. 2010; Fowler *et al*. 2015). Although a number of studies have reported a general positive effect of N enrichment on plants in N-limited forests and a negative effect of excess N (e.g., Aber *et al*. 1998; Högberg *et al*. 2006; Gilliam 2006; Thomas *et al*. 2010), specific responses of plants appeared to be highly growth form-dependent and ecosystem-dependent (LeBauer & Treseder 2008; Bedison & McNeil 2009; Dirnböck *et al*. 2014).

Due to the widespread heavy N deposition in Europe and America, numerous studies that focused on the growth responses of plants to N enrichment have been carried out in boreal and temperate forests during the past several decades (Magill 2000; Högberg *et al*. 2006). These studies showed that most trees have a positive growth response to N fertilization and therefore have higher potential carbon sequestration because the status of N limitation was largely alleviated by the increasing N inputs (e.g., Thomas *et al*. 2010; BassiriRad *et al*. 2015). However, the understory plants in these forest ecosystems inconsistently showed general negative responses to N enrichment with declined biomass or shifted community structure (Rainey *et al*.1999; Du *et al*. 2014; Dirnböck *et al*. 2014). In addition to the opposite responses of trees and understory plants to N enrichment, differences remained in the effects of N enrichment on single plant growth form in these forests. Generally, the limited light availability in these ecosystems with high tree canopy cover was ascribed to the negative effects of N fertilization (Strengbom & Nordin 2008).

Recently, the effects of N enrichment on tropical forests raised researchers' concern. Fertilization experiments in tropical forests showed different growth responses of trees to nutrient addition among individual size levels, understory shrubs and tree seedlings (Wright *et al*. 2011; Pasquini & Santiago 2012; Santiago *et al*. 2012) which contrasted with the ones

found for trees in the previously described experiments. For example, phosphorus (P)
fertilization enhanced the growths of small trees and seedlings but had no effect on median
and large trees, while N addition did not show any significant effect on plant growth in a
lowland tropical forest (Alvarez-Clare *et al*. 2013). In addition to the ubiquitous concept that
P was a critical element driving plant growth in tropical forests (Vitousek *et al.* 1991),
heterogeneous nutrient limitation that the growths of plants were co-limited by multiple
nutrients was further proposed to explain why diverse plants respond differently to N
enrichment (Wright *et al.* 2011; Alvarez-Clare *et al*. 2013; Wurzburger & Wright 2015).
Nevertheless, the patterns of specific nutrient limitation and responses of plants to N
enrichments among diverse forest ecosystems need further exploration.

As most of the nutrient fertilization experiments have focused on boreal forests, temperate
forests and lowland tropical forests, few studies have investigated the effects of N enrichment
on subtropical forests despite their broad distribution throughout the world and great
contribution to global carbon sink (Zhou *et al*. 2013; Yu *et al*. 2014; Huang *et al*. 2015).With
increasing N deposited in the subtropical ecosystems in southeastern China (Du *et al.* 2014),
it is important to diagnose the nutrient limitation and evaluate the responses of different plant
growth forms to N enrichment in subtropical forests for the assessment of carbon
sequestration and community dynamics.

To better predict the responses of subtropical forests and different plant growth forms to N
enrichment, we carried out a 3.4-year N fertilization experiment with three treatment levels
applied to nine 20 m $\times$ 20 m plots and replicated in three blocks in a subtropical forest in
south-eastern China. We attempt to explore whether N is a limiting element in the old-aged
evergreen broad-leaved subtropical forest. We hypothesize a positive response of trees to N
fertilization, but a negative response of understory growth forms to N fertilization due to the
expansion of canopy crown and consequent reduction of light availability.

**2 Materials and methods**

**2.1 Study site and experimental design**
The N fertilization experiment site was located at 30 $^\circ$01'47" N latitude and 117 $^\circ$21'23" E
longitude at an altitude of 375 metres in the natural conservation zone of Guniujiang in Anhui
Province, eastern China. As a commendable representative of the typical subtropical

broadleaved evergreen forest, the Guniujiang experimental site is an important part of the
NEECF (Network of Nutrient Enrichment Experiments in China's Forests) project (Du *et al*.
2013), because of its representativeness in both species composition and landscape structure
in the subtropical evergreen forest region. The study area has a humid climate with strong
summer monsoons with an annual average precipitation of 1,700 mm and an average annual
temperature of 14.9 ℃. The amount of wet N deposition in this region was 5.9-7.3 kg N
ha$^{-1}$ yr$^{-1}$. The soil in this area has been classified as yellow brown earth (Chinese Soil
Taxonomic Classification), and the $pH_{H2O}$ value at 0-10 cm soil depth was 4.58±0.05
(mean±SE). The total N, P, $NH_4^+$-N and $NO_3$-N content in the soil at 0-10 cm depth were 3.23
(0.37), 0.32 (0.02), 0.012 (0.001), and 0.002 (0.0006) mg g$^{-1}$, respectively (Li *et al*. 2015).

The study was conducted in a well-protected, mature subtropical evergreen forest (>300 year
age) with a three-layered vertical structure: the canopy tree layer (DBH>5 cm and height>5
m); the understory layer of saplings, shrubs and seedlings (DBH<5 cm and height<5 m); and
the ground-cover layer (ferns and herbs). The average density and basal area of trees were
1,219 trees ha$^{-1}$ and 36.35 m$^2$ ha$^{-1}$, respectively; *Castanopsis eyrei* was the dominant species
(which was also an important species at some other sites in subtropical forests) and accounted
for 87% of the total aboveground biomass of trees. The understory saplings and shrubs
contained several species, including *Cleyera japonica, Camellia cuspidata, Rhododendron
ovatum, Eurya muricata, Cinnamomum japonicum, Cinnamomum subavenium, Sarcandra
glabra,* and *C. eyrei*, and other native subtropical evergreen species (Table 1). Two fern
species (*Woodwardia japonica* and *Dryopteris hwangshanensis*) and an orchid (*Cymbidium
tortisepalum* var. *longibracteatum*) appeared on the floor layer, while *W. japonica* exclusively
dominated the floor layer with a coverage of 10%-20%.

We began N fertilization in March 2011. A randomized block design was used to avoid spatial
heterogeneity. We chose three blocks with similar stand growth, species composition and site
condition to establish three N treatments in each block: CK (0 kg N ha$^{-1}$ yr$^{-1}$), N50 (50 kg N
ha$^{-1}$ yr$^{-1}$), and N100 (100 kg N ha$^{-1}$ yr$^{-1}$). In total, nine 20 m ×20 m plots were established
with a 5-10 m buffer zone between each plot. The total $NH_4NO_3$ was divided into 12 dosages
and applied to the forest in each month at regular intervals. $NH_4NO_3$ in dosages of 0.48
kg/plot and 0.95 kg/plot were dissolved in 15 L of fresh water, respectively, and then sprayed
uniformly in N50 and N100 plots using a back-hatch sprayer. The unfertilized plots (controls)
were similarly treated with 15 L of fresh water without $NH_4NO_3$.

150

**2.2 Sampling and measurement**

In March 2011, the species of all trees higher than 2 m in each plot were labelled and their initial DBH (1.3 m) was measured. Then, autonomous band dendrometers made of aluminium tape and springs were installed on trees with a DBH greater than 5 cm. After one month to allow the tapes and springs on the trees to become stable, we began to measure the changes in the gaps on the tapes using vernier callipers (measured in July 2014) and then calculated tree DBH according to the following equation:

$$DBH = DBH_1 + \frac{X_2 - X_1}{3.14 \times 10}$$

where $DBH_1$ represents the initial DBH (cm) of trees measured in March 2011, and $X_2$ and $X_1$ (mm) represent the widths of gaps on the tapes measured in July 2014 and at the beginning of the experiment, respectively.

The basal area is a common indicator for weighing the biomass of trees. Therefore, tree basal area increments were calculated to indicate the responses of tree biomass to the N fertilization. First, to test community-level responses of tree layer to N fertilization, we calculated the sum of total basal area increase ($m^2$ $ha^{-2}$ $year^{-1}$) of all trees in a plot after 3.4 years of N fertilization and divided this value by the period of N fertilization (3.4 years) to obtain the annual basal area increase rate of the trees (dead trees were not included). Second, relative annual basal area growth rate (RGR, $m^2$ $m^{-2}$ $year^{-1}$) was used to eliminate the conceivable interferential effects resulting from the differences in the number and size of original individuals among plots according to the following equation, similar to Alvarez-Clare et al.'s method (2013):

$$RGR = \frac{\ln(2014\,BA) - \ln(2011\,BA)}{3.4}$$

where RGR represents the relative annual basal area growth rate ($m^2$ $m^{-2}$ $year^{-1}$), BA indicates the sum of basal area of all trees in each plot, and 3.4 (years) is the N fertilization period.

Because *C. eyrei* was the only dominant species in the tree layer, we separated it from other tree species and grouped its individuals into three classes based on their DBH values (i.e. 5-10 cm, 10-30 cm and >30 cm) to investigate the effects of N fertilization on the growth of trees after removing the plant species and original size factors. During the monitoring of tree growth, dead trees were recorded. Then, we calculated the aboveground biomass increments

of trees and the proportion of dead biomass using allometric equations (see Table S1).

We examined the effects of N fertilization on understory tree saplings distributed in the plots
according to their sizes and characteristics. For small trees with DBH<5 cm and height>2 m
(defined as "saplings"), DBH was measured at the beginning of N fertilization and in July
2014. Then, annual basal area growth rate and RGR of saplings were calculated based on
DBH changes. For very small trees or shrubs with DBH<5 cm and height<2 m (defined as
"shrubs/seedlings"), we set two 5 m × 5 m subplots in each plot along a diagonal direction
and investigated the abundance, dominance, basal diameter (diameter at 10 cm above the
ground), height and crown diameters of all shrubs/seedlings inside the subplots at two
specific times. The first time was at the beginning of N fertilization (March 2011), and the
second was in July 2014. The length, width and number of fern leaves were measured
carefully in the above-mentioned subplots, and the allometric equations for seven dominant
species were then obtained (Table S1). Because the average aboveground biomass of
shrubs/seedlings and ferns showed no significant differences across three N treatments before
N fertilization in March 2011, we regarded the distribution of these understory
shrubs/seedlings and ferns to be homogeneous among the three treatments. Then we
identified the effects of N fertilization by comparing the aboveground biomass of
shrubs/seedlings and ferns in 2014 among the different treatments. Meanwhile, to investigate
the canopy cover and understory light availability, we used a digital camera (Canon, Japan)
with a fisheye lens (Sigma circular fisheye) to take photographs of canopy. In each subplot,
we put the camera at 1m above ground and took 5 photos upwards from understory.

In addition, to further explore the influences of N fertilization on plants' growth from
biogeochemical aspect, we measured soil N, P content and pH. Specifically, we set three
subplots randomly within each plot and collected three subsamples of 0-10 cm soil for each
subplot using a hand-held steel soil borer (3 cm in diameter), during investigation of the
understory plants. Then, the three subsamples were mixed together to form one sample per
plot and transported to a laboratory and air dried naturally. After air-dried, soil samples were
ground with a ball mill (NM200, Retsch, Haan, Germany) and screened through a 100 mesh
sieve. The N concentration of soil was measured using an elemental analyser (2400 Series2
CHNS/O Elemental Analyzer, Perkin-Elmer, USA). After acid digestion of the samples, soil P
concentrations were measured using a flow injection analysis instrument (AutoAnanlyzer3,
Bran+Lubbe City, Germany). Soil pH was measured by dry soil in water suspension with a
water:soil ratio of 1:2.5.

**2.3 Statistical analysis**
We used an analysis of variance (ANOVA) to evaluate the effects of N fertilization on soil N
and P content, soil pH, tree basal area increments, RGR, aboveground biomass increments,
proportion of dead trees, and aboveground biomass of shrubs/seedlings and ferns. Block and
N treatment were both regarded as fixed factors in the statistical model. We excluded the
interactions between block and N treatment from the model because they do not have
ecological meaning. Tukey's honest significant difference (HSD) tests were used to conduct
the multi-comparisons among the three N treatments. For the estimation of canopy cover, we
followed the detailed procedures of weighted ellipsoidal method using the software of
Hemisfer (version 2.16.6) to obtain values of vertical total gap fraction (Fmv) which indicate
the proportion of projected light spots to the total projected area (Thimonier *et al*.2010).
Then we obtained the values of [1-Fmv] to indicate canopy cover. All statistical analyses
were performed in R.3.2 (R Development Core Team, 2010), and all figures were drawn in
SigmaPlot 12 (Systat, 2010).

**3 Results**

**3.1 Effects of N fertilization on canopy cover, soil N and P contents and pH**
The indicator of forest canopy (i.e. [1-Fmv]) showed no significant differences between
unfertilized and fertilized plots with 3.4 years of N fertilization (Table 2). Although the fish
eye measurements did not provide evidence for the changes in total forest cover with the
effects of N fertilization, there still may be a shift between the contribution of overstory and
understory trees to the total forest cover.

3.4 years of N fertilization significantly increased the N content of 0-10 cm soil ($p$=0.03),
especially in N100 plots (Fig. 1a), but showed no significant effect on soil P content (Fig. 1b,
$p$>0.05), thus leading to a significant increase in soil N:P ratio (Fig. 1c, $p$=0.02). Additionally,
the N fertilization also decreased soil pH and aggravated soil acidification (Fig. 1d, $p$=0.05).

**3.2 Growth responses of trees to N fertilization**
The increments of absolute basal area, aboveground biomass and RGR of all trees at plot
level showed no significant response to N fertilization during 3.4-year N fertilization (Fig.
2a~2c). Compared with the unfertilized plots, N50 and N100 fertilized plots showed a
tendency toward higher averaged proportions of dead trees' aboveground biomass despite no
significant difference between them (Fig. 2d).

Individuals of the dominant species *C. eyrei* with different initial DBH showed divergent
responses of absolute basal area increments and RGR to N fertilization (Fig. 3a-3f). The
small trees with a DBH of 5-10 cm growing under unfertilized plots showed greater basal
area increments than those growing under N fertilized plots (Fig. 3a, *p*=0.02). Specifically,
the N50 and N100 fertilization decreased the absolute basal area increments of small
individual trees at rates of 2.2 $cm^2\,tree^{-1}\,year^{-1}$ and 1.98 $cm^{-2}\,tree^{-1}\,year^{-1}$, respectively, which
indicated that the decreasing degrees of the absolute basal area of small trees reached 66.4%
and 59.5% in N50 and N100 plots. The small individual trees also showed a tendency toward
lower averaged RGR in N fertilized plots although no significant difference was detected
between them (Fig. 3d, *p>0.05*). As opposed to the negative responses of small trees to N
fertilization, the basal area increment and RGR of median *C. eyrei* individuals (DBH of 10-30
cm) and large *C. eyrei* individuals (DBH of >30cm) showed no significant response to N
fertilization, but the averaged growth rate of large *C. eyrei* individuals in N50 plots almost
doubled the value of the corresponding large individuals in unfertilized plots (Fig. 3b-3c and
3e-3f, *p>0.05* in all cases).

**3.3 Growth responses of understory saplings, shrubs/seedlings, and ferns to N**
**fertilization**
Responses of understory saplings to N fertilization were similar to those of small dominant
trees. Although the annual absolute increments of basal area increments of saplings showed
no significant response to N fertilization (Fig. 4a, *p>0.05*), the RGR of sapling growing in
N50 and N100 plots showed a substantial decrease at rates of 0.021 $m^2\,m^{-2}\,yr^{-1}$ and 0.019 $m^2$
$m^{-2}\,yr^{-1}$, respectively, compared to sapling growing in unfertilized plots (Fig. 4b, *p<0.001*). In
addition, a general negative effect of N fertilization also occurred on understory shrubs and
ground-cover ferns. The aboveground biomass of seven predominant shrubs/seedlings was
drastically decreased by 69.4% and 79.1% in N50 and N100 fertilized plots, respectively,
compared with those in the unfertilized plots (Fig. 5a, *p<0.01*). Remarkably, the aboveground
biomass of ground-cover ferns significantly declined by 92.4% and 93.4% in N50 and N100
fertilized plots (Fig. 5b, *p<0.05*).

**4 Discussion**

**4.1 Growth responses of trees to N fertilization**

Nutrient limitation was generally determined through evaluating ecosystem feedbacks to nutrient addition (Vitousek 1991; Santiago *et a.* 2012; Alvarez-Clare *et al.* 2013). When the forest ecosystems showed a positive response to added nutrient, e.g., plant growth or rates of physiological processes were promoted, the added nutrient then could be interpreted as limiting to the ecosystem, otherwise, as not limiting to the ecosystem (Santiago 2015). We initially expected positive growth responses of trees exposed to N fertilization in this subtropical forest because N availability in the soil would be enhanced by N fertilization and the potential N limitation of plants in the forest ecosystem could be alleviated. However, contrary to our expectation, we did not observe strong positive growth responses of trees to N fertilization (Figs. 2 and 3). Across individual trees of different sizes and plant growth forms, we only observed substantial negative responses of small trees (5-10 cm DBH; Fig. 3a and 3d) and saplings (Fig. 4a and 4b) and weak responses of median and large trees (>10 cm DBH) to N fertilization (Fig. 3b-3c and 3e-3f), which further demonstrated that the growth of trees in this old-aged subtropical forest was not essentially limited by N as hypothesized.

Contrasted with previous positive responses of trees to N fertilization in boreal and temperate forests which were considered as N limited ecosystems (Högberg *et al.* 2006; Thomas *et al.* 2010; BassiriRad *et al.* 2015), our finding of the unchanged responses of trees to N fertilization was partly consistent with observations of trees from tropical forests (e.g., Santiago *et al.* 2012; Alvarez-Clare *et al.* 2013). Studies from mature tropical forests have revealed that P availability was a critical element shaping tree species distribution and productivity (Santiago 2016; Dalling *et al.* 2016). Given the similar high-weathered soil properties, humid climatic conditions and dominant evergreen broadleaf trees in mature subtropical forest as those in wet tropical forest, we speculated that P limitation, rather than N limitation, might have played a key role in influencing growth of plants in subtropical forest.

The N and P stoichiometry of soil might have objectively provided indicators of P limitation with the effects of N fertilization in this subtropical forest, because soil N contents and N:P ratio in N fertilized plots were remarkably higher than those in unfertilized plots (Fig. 1). Additionally, limitation of other nutrients, such as K (potassium) which was highlighted in tropical forests, and their combination as well as heterogeneous nutrient limitation of specific species, plant growth forms and individuals in different sizes may warrant further

consideration in subtropical forests (Wright *et al.* 2011; Santiago *et al.* 2012; Alvarez-Clare *et al.* 2013).

Moreover, the high spatial heterogeneity in old-aged subtropical forest, similar to tropical forests, could be a possible explanation for the lack of significant responses of plot-averaged basal area growth, RGR, aboveground biomass of trees with a DBH of >5 cm and the proportion of dead trees to N fertilization. In eastern China, the distributions of subtropical forest stands are quite topographically fragmented, while relative flat stands are required to avoid N losses and minimize spatial heterogeneity among experimental treatments. The actual distribution and topography of the subtropical forests limited the number of replications in the N fertilization experiment. This limitation might reduce the statistic power of N treatment on plot-averaged plant growth rate which has been pointed out in previous studies (Wright *et al.* 2011; Alvarez-Clare *et al.* 2013). Furthermore, our observation of large trees with DBH >30 cm showed that the averaged growth rate of large *C. eyrei* individuals in N50 plots almost doubled the value of the corresponding large individuals in unfertilized plots. Nevertheless, the results of ANOVA showed that the effect was not significant. As the number of large trees in the experiment was relatively less than the small trees, the low replication and high spatial site heterogeneity might have reduced the statistical power of N fertilization on the large trees. Thus, fertilization experiments with more homogeneous plots and more replicates are warranted to further strengthen these findings. Overall, given the negative and potential positive effects of N fertilization on small and large trees, it is of urgent necessity to conduct long-term monitoring of the trees which would provide alternatives for accurately evaluating the forest dynamics under the enhanced global N deposition.

**4.2 Growth responses of small trees, understory saplings, shrubs/seedlings and ferns to N fertilization**

Although the positive responses of small or juvenile trees to nutrient fertilization has been reported in boreal, temperate and tropical forest (e.g., H ögberg *et al.* 2006; Bedison & McNeil 2009; Alvarez-Clare *et al.* 2013), our results showed a remarkable negative effect of N fertilization on small-sized plants including trees, understory saplings, shrubs/seedlings and ferns. During our field investigation, we also found that the average proportion of dead trees (Fig. 2d) tended to increase in N fertilized plots although the result was not statistically significant ($p = 0.50$). Additionally, the ground-cover ferns in N100 plots almost disappeared

after 3.4-year N fertilization (personal observation). Given the high stand density in this mature subtropical forest, we suggest that N fertilization might potentially lead to increased self- and alien-thinning of individuals through decreasing understory light availability.

The pivotal role of light availability in the eco-physiological processes of understory growth forms has been widely recognized (Santiago 2015). Due to the limited light availability, understory plants may not be able to incorporate the added nutrient and promote their photosynthetic rates (Alvarez-Clare *et al*. 2013). However, a study conducted in tropical forest with thick canopy showed that photosynthetic process could be enhanced by nutrient addition even under low light availability (Pasquini & Santiago 2012). In a sharp contrast, the study conducted in an Australian rainforest revealed that understory seedlings increased growth when the light availability was high, but showed no significant response to nutrient fertilization in low lights (Thompson *et al.* 1988). These studies, together with our field observations, suggest that the growth of understory plants is largely co-limited by nutrient and light availability in the local environment. Further, our results of forest canopy cover estimated by photographic fisheye showed no significant differences between unfertilized and N fertilized plots, which was consistent with the findings of Lu et al. (2010). Although the understory light irradiance fluctuated largely during a day and was very hard to detect precisely, our measurements of forest canopy cover provided a rough evaluation for light availability and a potential shift between the contribution of overstory and understory trees to the total forest cover which could partly explain the differences in the responses of trees with different sizes (i.e. different DBH classes). The results might indicate that other factors in addition to the low light availability in this old-aged forest had also played a crucial role in influencing understory plants during 3.4 years' N fertilization.

**4.3 Potential N saturation and plant growth**

The striking biomass reduction of the understory plants, especially ferns, in response to N fertilization in our study well corroborated the similar findings in an old-aged tropical forest at Mt. Dinghushan in China (Lu et al., 2010). Also, consistent with previous studies obtained from boreal, temperate and tropical forests (Rainey *et al.* 1999; Alvarez-Clare *et al.* 2013; Dirnböck *et al.* 2014), our experiment revealed that understory small-sized plants responded sensitively to nutrient fertilization, which might indicate a possibility of N saturation in the subtropical forest. According to the definition of N saturation addressed by Aber et al. (1998)

(i.e. N availability in the forest ecosystem exceeded the demand of plants and microbes), the
drastic decrease of understory ferns, shifted composition of understory plant community, and
cation imbalances of understory species after 7 years' chronic N fertilization at Harvard
Forest, USA, could be interpreted as useful indicators of N saturation (Rainey *et al.* 1999).
Moreover, a 6-year N fertilization experiment in an old-aged tropical forest at Mt.
Dinghushan also showed signs of N saturation, such as significant increases in nitrate ($NO_3$-)
leaching, inorganic N concentration and $N_2O$ emissions of soils, and soil acidification (Lu *et*
*al.* 2014; Chen *et al.* 2015). In our experiment, the soil acidification and increased soil N
concentration in high N fertilized plots (Fig. 1) combined with the negative responses of
understory plants suggest that the 3.4-year N fertilization in this mature subtropical forest site
has potentially caused N saturation. Nevertheless, further observations are still required to
explore the mechanisms underlying the changes of different growth forms with the effects of
N enhancement in the subtropical forests.

**5. Conclusion**

Contrasting growth responses among plant growth forms to N fertilization were present in the
mature subtropical evergreen forest in this study. Overall growth of trees at the plot level
showed no significant response to the N fertilization; however, if the dominant tree species *C.*
*eyrei* was grouped into three DBH classes, the basal area increment of small trees with a
DBH of 5-10 cm declined 66.4% and 59.5% in N50 and N100 fertilized plots, respectively,
while the growth of median and large trees with a DBH of >10 cm showed weak responses to
N fertilization. The growths of understory saplings, shrubs/seedlings, and ground-cover ferns
showed a negative response to N fertilization. Our results indicated that N might not be a
limited nutrient in this subtropical forest and that other nutrient and light availability may
potentially co-limit growth of plants with different growth forms. Our data also suggested
that even short-term N fertilization might have caused N saturation in this mature subtropical
forest and the limitation of other nutrients might be amplified with increasing N addition.
*Funding:* This study was funded by the National Natural Science Foundation of China
(31321061 and 31330012).

*Acknowledgements:* We wish to thank Bernhard Schmid, Gianalberto Losapio, Lilian Dutoit,
Peter Schmid and Jessica Baby for their helpful suggestions on the manuscript, and the editor
and two anonymous reviewers for their insightful comments that greatly improved this
manuscript. We also thank the Sino-German Center for Research Promotion for the
participation in a summer school in Jingdezhen (GZ1146).

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

**Table 1** Growth measurements for four plant growth forms in this study before N fertilization.
Numbers in the tables represent means (or mean $\pm$ (standard error), $n$=9) of plants across all
plots. TBA: total basal area of trees; DBH: diameter at breast height (1.3 m); Basal diameter:
diameter at 10 cm above the ground.

| Growth forms | Species | Growth variable | | |
|---|---|---|---|---|
| | | TBA ($m^2 ha^{-1}$) | DBH (cm) | Height (m) |
| Trees | *Castanopsis eyrei* | 32.5 $\pm$2.7 | 15.7 $\pm$3.6 | 11.8 $\pm$2.1 |
| Saplings | *C. eyrei* | 0.61$\pm$0.10 | 3.81 $\pm$0.04 | 2.59 $\pm$0.06 |
| | | Coverage (%) | Basal diameter (mm) | Height (cm) |
| Shrubs & Seedlings | *Cleyera japonica* | 2.89 | 9.24 $\pm$5.13 | 79.8 $\pm$40.82 |
| | *Camellia cuspidata* | 8.60 | 7.01 $\pm$0.62 | 60.1 $\pm$4.37 |
| | *Rhododendron ovatum* | 5.97 | 16.81$\pm$8.91 | 167.5 $\pm$65.02 |
| | *Eurya muricata* | 3.04 | 7.00 $\pm$1.57 | 111.0 $\pm$38.16 |
| | *Cinnamomum japonicum* | 2.85 | 4.44 $\pm$1.46 | 51.1 $\pm$26.59 |
| | *Cinnamomum subavenium* | 5.03 | 2.77 $\pm$0.64 | 29.9 $\pm$7.54 |
| | *Sarcandra glabra* | 2.92 | 3.60 $\pm$0.11 | 35.7 $\pm$3.69 |
| | | Density (shoots $m^{-2}$) | | |
| Ferns | *Woodwardia japonica* | 1.19 $\pm$0.23 | | |


Table 2 The indicator of canopy cover (i.e. [1-Fmv]) of the three treatments in our
experiments. *n* indicates the number of replicates. *SE* indicates the standard error.

| Treatment | *n* | Canopy cover | |
|---|---|---|---|
| | | mean | SE |
| CK | 3 | 0.77 | 0.01 |
| N50 | 3 | 0.76 | 0.04 |
| N100 | 3 | 0.72 | 0.01 |


**Figure 1** Effects of N fertilization on soil nutrient content, N:P ratio and pH (mean ±se, n=3)
at the soil depth of 0-10 cm. (a) Total N content per gram soil; (b) total P content per gram
soil; (c) N:P ratio and (d) soil pH. Numbers in these figures indicate the results of ANOVA .

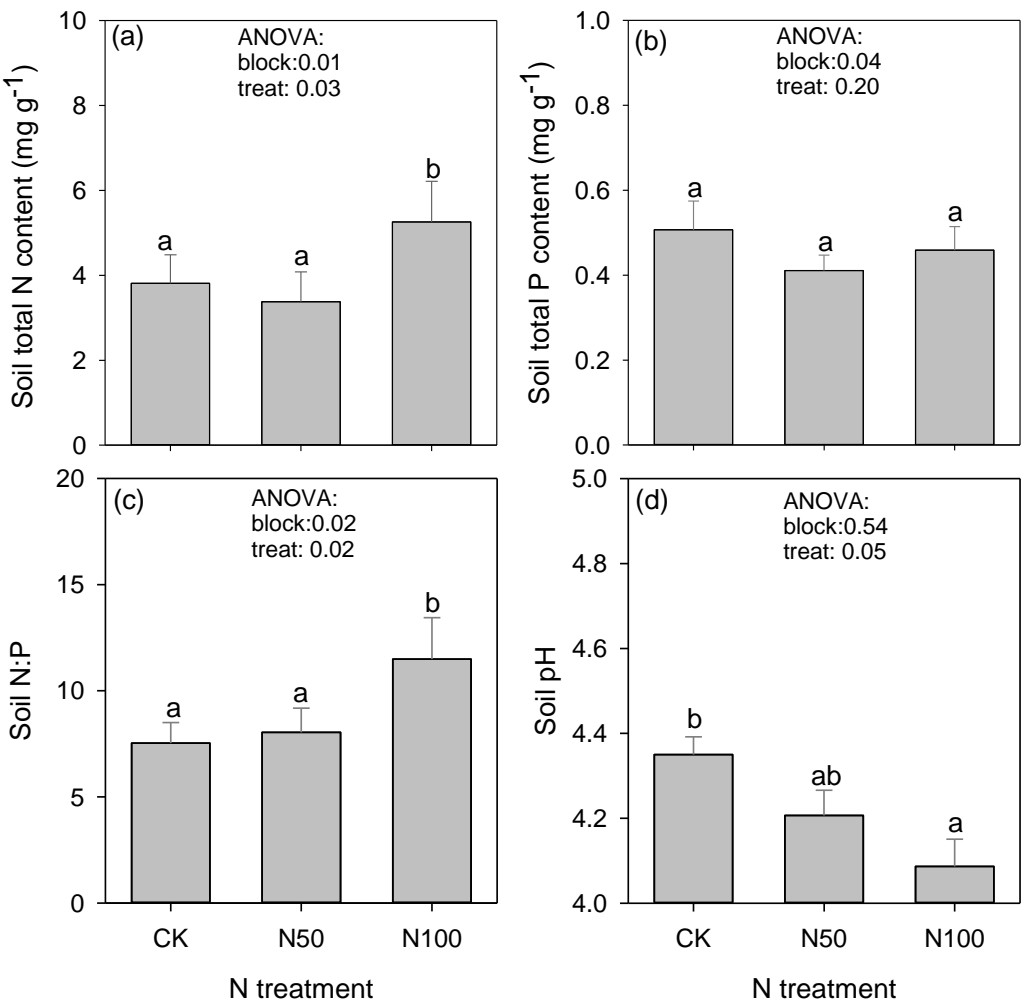


**Figure 2** Effects of N fertilization on the growth and mortality of all trees (mean ±se). (a)
Absolute basal area increase of all trees; (b) aboveground biomass increase of all trees; (c)
relative growth rate of total tree basal area; and (d) the proportion of all dead trees. The
proportion of dead trees was calculated using the aboveground biomass of all dead trees
during the experiment divided by the total aboveground biomass of all trees in 2014.
Numbers in these figures indicate the results of ANOVA. The N treatment on x-axis
represents three levels of N fertilization: CK (0 kg N ha$^{-1}$ yr$^{-1}$), N50 (50 kg N ha$^{-1}$ yr$^{-1}$) and
N100 (100 kg N ha$^{-1}$ yr$^{-1}$).

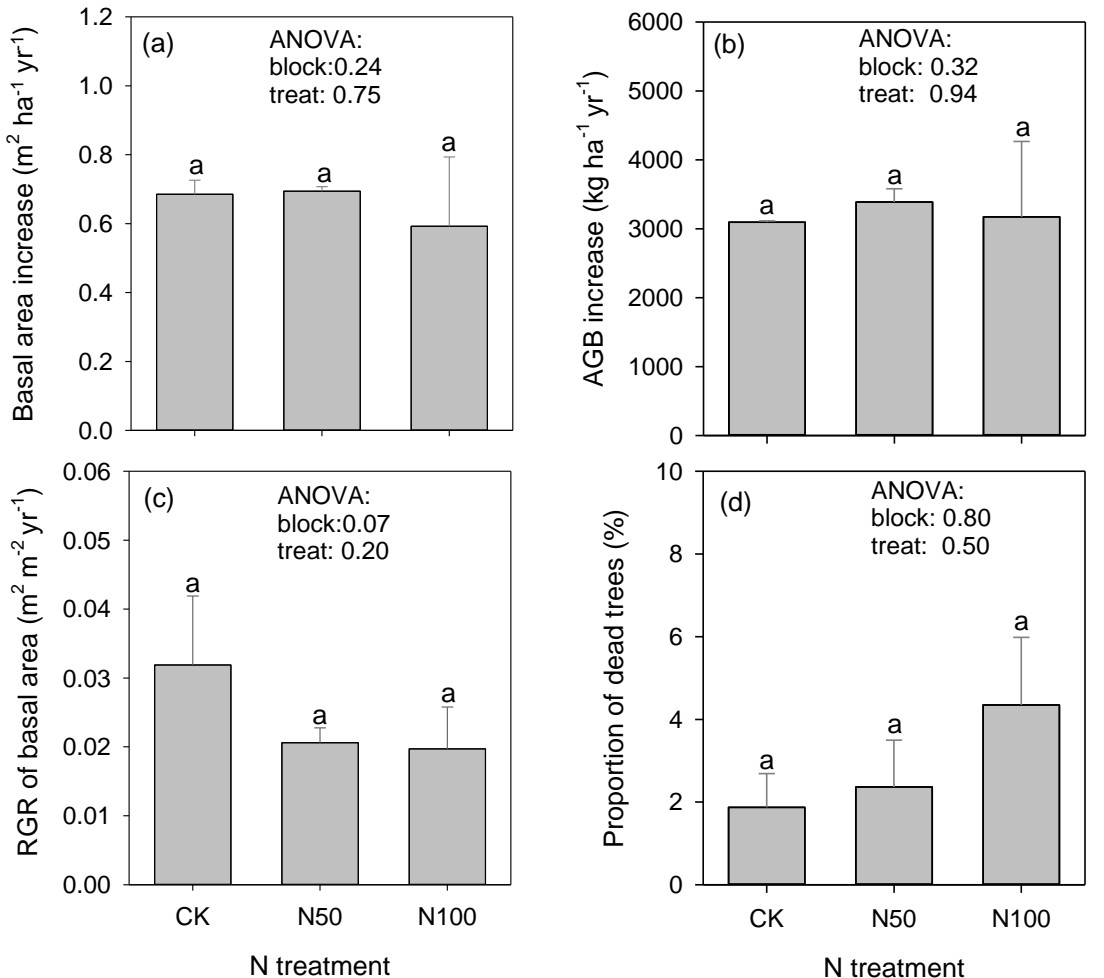


**Figure 3** Effects of N fertilization on the growth (mean ± se) of *C. eyrei* by DBH classes
(5-10 cm, 10-30 cm and >30 cm). (a-c) Absolute basal area increase and (d-f) relative growth
increase rate of basal area. Numbers in these figures indicate the results of ANOVA. The N
treatment on x-axis represents three levels of N fertilization: CK (0 kg N ha$^{-1}$ yr$^{-1}$), N50 (50
kg N ha$^{-1}$ yr$^{-1}$), and N100 (100 kg N ha$^{-1}$ yr$^{-1}$).

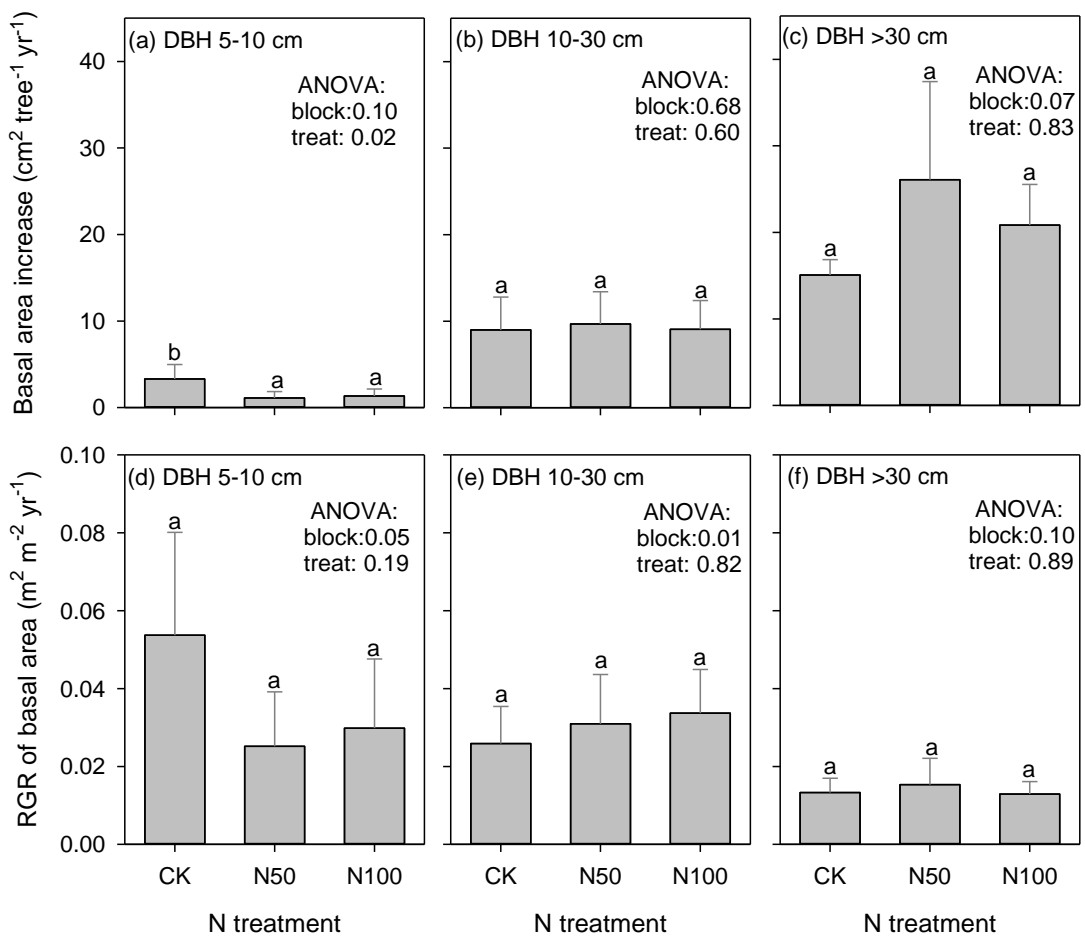


**Figure 4** Effects of N fertilization on the growth of saplings (mean ± se). (a) Absolute basal
area increase and (b) the relative growth rate of basal area. Numbers in these figures indicate
the results of ANOVA. The N treatment on x-axis represents three levels of N fertilization:
CK (0 kg N ha$^{-1}$ yr$^{-1}$), N50 (50 kg N ha$^{-1}$ yr$^{-1}$) and N100 (100 kg N ha$^{-1}$ yr$^{-1}$).

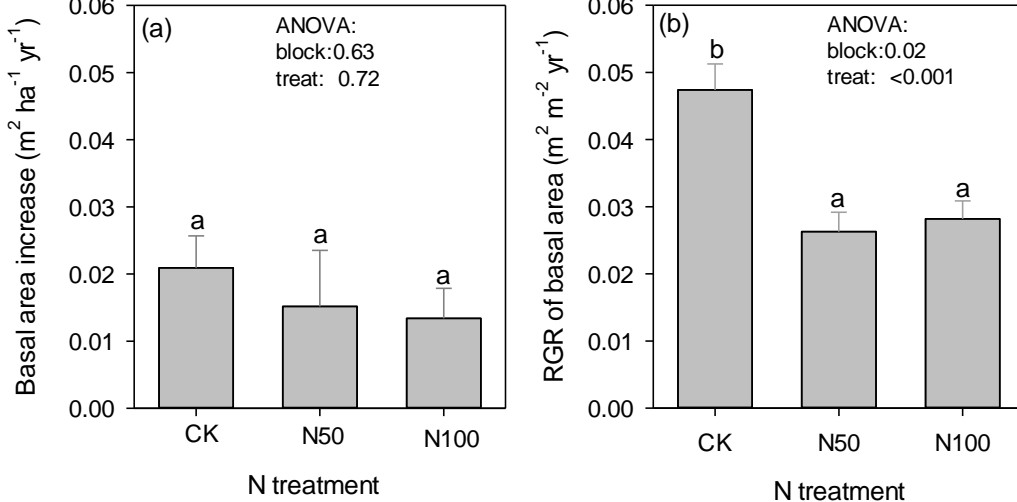


**Figure 5** Effects of N fertilization on the aboveground biomass of shrubs, seedlings and ferns.
Bars show the aboveground biomass of (a) shrubs/seedlings and (b) ferns (mean ±se).
Numbers in these figures indicate the results of ANOVA. The N treatment on x-axis
represents three levels of N fertilization: CK (0 kg N ha$^{-1}$ yr$^{-1}$), N50 (50 kg N ha$^{-1}$ yr$^{-1}$) and
N100 (100 kg N ha$^{-1}$ yr$^{-1}$).

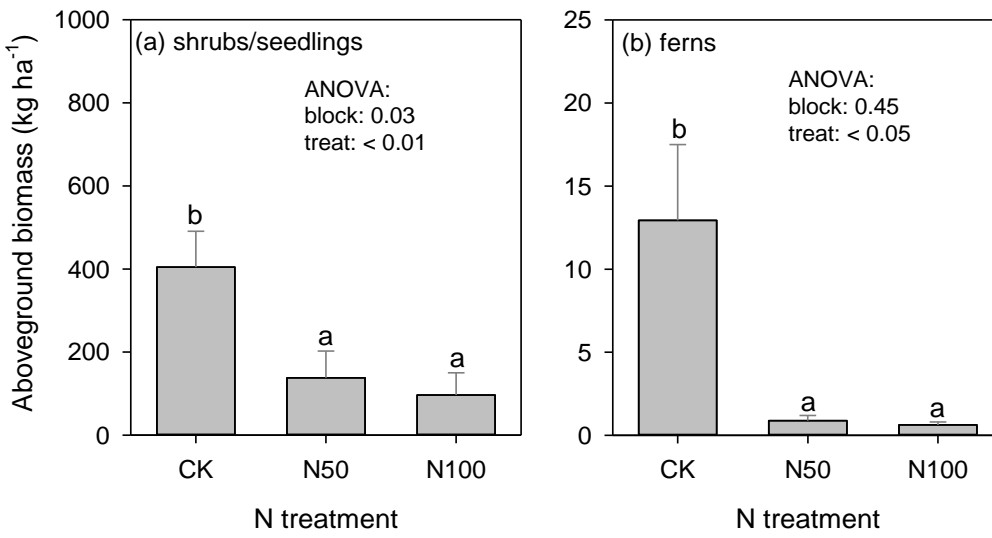
