# Peer review of "Growth responses of trees and understory plants to nitrogen fertilization in a subtropical forest in China"

_Biogeosciences, 2016_

## Referee Comment (RC1) · Anonymous Referee #1 · 28 Nov 2016

This paper describes the results of a 3 year (authors say 4 in the abstract) forest N fertilization study conducted in China. The study focuses on growth of trees, saplings, shrubs and understory growth and mortality. The authors main conclusion is that N fertilization affects the various plant growth forms in different ways, with the smaller plants being most affected. Overall, this paper adds to the growing knowledge regarding N impacts on forest ecosystems, but suffers from many of the limitations that other fertilizer studies have to deal with 1) environmental relevance of the dosage amount and form, 2) short (3 year) period for assessment and 3) no data to support the mechanisms of the observed impact. Further, the study has low replication (3 20 x 20 m plots) per treatment. My suggestion is that in the revised paper - these limitations should be fully

addressed and evaluated with respect to the implications for the overall conclusions made by the authors.

Specific comments

1. Environmental relevance - The application rates of 50 and 100 kg/N/ha are very high and I suspect are found in a few locations in China, but not likely widespread. My experience is that such high dosages almost always produce some effect but 1 year of 50 kg/N/ha is not the same as 5 years at 10 kg/N/ha. The authors should read a very good paper by Lovett and Goodale (2011) Ecosystems - that discusses this issue. Further, if my math is correct the authors are applying 100 kg N in 12 dosages per year, each time in 15 L of water. This makes 8kg N per time - dissolved in 15L, which is about 440g/L. Given the reportedly greater impacts of the treatments on the ground species, I am wondering about the direct effects of this spray? This should be discussed/evaluated.

2. This is a short study (3.4 years - should be consistent throughout which it isn't at present) with relatively low replication. In both instances real changes may be occurring but statistically they are not different among treatments. Throughout the paper the authors refer to differences among treatments - when in fact they are not significant (e.g. Figure 3). Over time or with more replication it could be true - just as equally it may still be noise in the system. The authors are guilty of talking about differences when in fact they statistically the same.

3. The main argument for the difference in response among growth forms is shading. There is no evidence for this presented in the manuscript (not measured). Equally, there is no evidence for statistical differences in N content among treatments (supplementary info). Thus while the authors present a mechanistic reason behind the differences there is no real statistical evidence to support these claims. Changes in canopy cover were not assessed and N or P (nothing else shown) are not significant among treatments. Soil pH is lower, but Al or Mn are not measured. I found the discussion section (4.3) very misleading for example - "total N content of soil was enhanced by N fertilization and P concentration in plant leaves and in fine roots showed that N concentration increased" - not only is this a poor sentence, it is factually incorrect - N content did not increase in the 50 Kg N treatment nor did N content significantly increase (Figures are actually labeled incorrectly). Similarly there is no evidence of P being lowered by the treatment (soil or plant). Why was nitrate or ammonium not measured?

4. The P fertilizer study added at the end reads just like an add on and does not help the paper and it should be deleted. Similarly the text on lines 243-249 could be deleted.

5. The data shown in Figure 2 - basal area changes over time by size class are self-evident and this could be deleted. I am much more interested in how size class distribution compared among the study plots at the beginning of the study period. With such low replication (~40 trees per plot = 120 trees per treatment, which then get broken down into smaller units - some of these comparisons may be being made on a very few trees).

As addressing these comments should alter the paper substantially I will not comment on editorial issues.

---

## Referee Comment (RC2) · Anonymous Referee #2 · 16 Jan 2017

The paper describes how a subtropical forest system responded to three levels of N addition (0, 50 and 100 kg N ha-1 year-1). The authors describe the response for over and understory trees, saplings, shrubs and understory vegetation. In general the responsiveness of the system was limited, but larger overstory trees responded by increased growth (basal area), while other parts of the vegetation showed no increase in growth, or even showed suppressed growth. The main conclusion made is that different components of the vegetation respond differently to N addition and that this should be considered when effects of anthropogenic N deposition is evaluated.

Major comments

My opinion is that the text in large parts of the paper needs to be rephrased. The

results needs to be much more carefully described and the authors should make an effort in making it more clear what differences that are statistically supported and what are not. Several of the main results discussed (e.g. that N addition stimulated growth of large trees and suppressed growth by small ones) is not supported by data. I agree that it is likely that the suppression of understory vegetation stems from increased light competition with a denser overstory, and that this was caused by N addition, But this is NOT reflected by any of the data collected by the authors. Perhaps, N addition increased leaf area or canopy cover of the overstory, and by this suppressed light conditions and the growth of the understory? Such effect would over time be expected to be reflected by increased basal area but the limited duration of the experiment (3 and not 4-year as claimed in the text) may have been too short the capture such response. If there are any data on canopy cover or light transmission to the ground level, such data would definitely be worth exploring as it may help explaining the results.

The addressed questions could easily be made a bit more sophisticated by asking for differences compared to the known response from other forest systems (e.g. temperate, tropical or boreal forests). This is partly related to how the available knowledge from other systems is described in the introduction (see comments further down).

The last part of the abstract can be misleading as the result presented only supports that small trees grow better under ambient N than elevated N and there is actually no support at all for higher growth of large trees under elevated N.

The last sentence of the abstract, i.e. the conclusion of the study/implication of the results is extremely vague as the reader is not provided with any clue to why it is important to consider more parts of the vegetation than just the trees. A hint may be given by the results presented, i.e. that large trees responded differently from other parts of the vegetation, but the authors never help the reader describing why this is problematic of what can happen if the response is just evaluated based on the trees (large trees).

I miss information on whether the growth in study system in general is N limited. In my opinion this is essential information when it comes to evaluating the response to the N addition. If not, or if the growth in the system is co-limited by other nutrients, a lack of N response should be interpreted a bit differently than if N is the solely or main limiting nutrient. I believe that this is important as the response to the N treatment in general was rather weak and most often non-significant. In fact the additional data presented on P addition (Fig. 6) might suggest that P is co-limiting nutrient.

L. 43-53. The authors seriously exaggerates the lack of knowledge, and I would go so far as saying that the content of this paragraph gives a false picture the available literature on N effects in forested systems. First, studies from boreal areas are not at all limited to tree response. In fact there has been much other work done, both on other plant groups and on other organisms than plants. For a quick overview see the summary paper by Bobbink et al 2010 (that is cited elsewhere in the ms). Second, the authors claim that the response of forest understory communities rarely have been studied, which is simply not true. Just a few examples are van Dobben et al. 1999 (For Ecol Manag 114, 83–95); Strengbom et al 2001 (Funct Ecol 15, 451–457); Gilliam 2006 (Journal of Ecology 94: 1176–1191), and there are many more.

L. 110-118. I can understand why you exclude trees that died, and understand why trees that had decreasing DBH were excluded (but not necessarily agree that they should be excluded, as you then only accept measuring errors in one direction but not the other), but how can you justify excluding trees that showed no change in DBH? I am very worried that by omitting trees that showed no change in DBH may have seriously have influenced the results of your study and risk exaggerating the positive response that the N addition may have had.

The authors should in general be much more careful when presenting non-significant differences. If these at all should be mentioned it should be absolutely clear to the reader that these are non-significant differences. Much of the discussion, and even parts of the major conclusions, deals with non-significant differences that are presented

as if they were statistically supported (e.g. L. 204-205, 210-212, 252-256).

Specific comments

L. 22-23. There was no response at all of the larger trees! Avoid bringing up differences that are far from significant in the abstract. This is not just wrong, it is misleading!

There is no description on how, when and why P was added in some plots.

L. 139-140. Do you have pre-treatment measures supporting that the vegetation was homogenous among plots at the initiation of the experiment? If so present these in a simple form. If not you should describe how the homogeneity was assessed.

Table 1. It is not clear what the data represents. Are the numbers presented grand mean across all treatments? If so this should be clearly stated in the text explaining the table.

L. 157-158. I do not understand the results described here "The basal area and RGR of trees at the community level showed no significant response to N fertilization (Fig. 1); however, the increase rates of basal area were likely hindered by N fertilization (Fig. 1c)"What does this mean? As far as I can see from the statistical results presented and the data presented in Fig 1 there is just simply a lack of N response. Very unclear what you mean when saying that growth was hindered by N addition?

L. 161-163. Be more careful when presenting non-significant differences. There might be a tendency towards more dead biomass under N addition but the difference is far from significant.

L. 164-165. The text here is wrong here. This result has nothing to do with the N treatment. The test and the fig just describes that basal area and RGR differed depending on size among individual trees of this species. This is very important as it seems like part of the conclusion is based on that there is a N effect here.

L. 168-173. The text here is in most parts misleading. The only effects that are supported by the data presented is that the smallest trees growing under no N addition had higher basal area and higher RGR than small trees growing under N addition. All other differences that may or may not be visible in the figure is far from statistically supported and should not be mentioned here in the results.

L. 175-179. Is the test result presented in the fig correct? According the test results N addition influences RGR and mortality, but form the post hoc test there seem to be difference among the groups. From inspecting the data presented in the fig I wonder if there is some error among the letters indicating the differences among the groups in panel c and d.

Results covering the data presented in fig 6 is missing from the result section

L. 192-194. What is the rationale for expecting a common positive response for all types of plants? To me this seems a bit naïve, given that forest plant communities often are size structured communities (see e.g. papers by Peter Grubb), and understory species than can be expected to be light rather than nutrient limited.

L. 204-205. The first part of the sentence (large trees) is NOT supported by the results.

Minor technical and language errors

The text is in need of some language edition. I just provide a few examples were the text need some re-phrasing. I have not paid that much attention to text editing as I believe that the paper need to be substantially revised before the paper can reach an acceptable standard.

L 21-22. …the small trees with DBH (diameter at breast height) values of 5-10 cm were hindered by N fertilization…In what way was the small trees hindered?

L. 23-24 … Small trees, saplings and particularly understory shrubs and ground-cover ferns suppressed seriously by increasing N fertilization… How are the suppressed? I am not very fond of the wording seriously as it is not a neutral wording. Better describe how large the difference was.

L. 24-24. ...Proportion of mortality? Here it is better to write either the mortality of plants were... or The proportion of plants that died...

L 177. Avoid evaluating your results in the result section by using wording such as severely here. Save that type of wording for the discussion.

L. 180-185. There is no need to present mean values in text if these are shown in the figure 5. Do not present data twice, choose either to present then in text or in the fig.

---

## Author Comment (AC1) · 7 Feb 2017

Authors' response to reviewer' comments on the manuscript bg-2016-416 "*Contrasting growth responses among plant growth forms to nitrogen fertilization in a subtropical forest in China*" by Di Tian et al.

Di Tian, tiandi@pku.edu.cn
Jingyun Fang, jyfang@urban.pku.edu.cn

**To the editor:**

Dear Dr. Zaehle,

We appreciate your help very much in developing the manuscript and your devotion to find suitable referees. Also, we appreciate the comments from two anonymous referees. The major comments were focused on the design of N fertilization including N dosages and limited replications, and unclear description of our results, as stated in a separated letter we have wrote to you. We have carefully studied the comments and rephrased the introduction, results and discussion in the updated version. The point-by-point responses are as follows.

**To Anonymous Referee #1:**

[**Major Comments**]: This paper describes the results of a 3 year (authors say 4 in the abstract) forest N fertilization study conducted in China. The study focuses on growth of trees, saplings, shrubs and understory growth and mortality. The authors main conclusion is that N fertilization affects the various plant growth forms in different ways, with the smaller plants being most affected. Overall, this paper adds to the growing knowledge regarding N impacts on forest ecosystems, but suffers from many of the limitations that other fertilizer studies have to deal with 1) environmental relevance of the dosage amount and form, 2) short (3 year) period for assessment and 3) no data to support the mechanisms of the observed impact. Further, the study has low replication (3 20 x 20 m plots) per treatment. My suggestion is that in the revised paper - these limitations should be fully addressed and evaluated with respect to the implications for the overall conclusions made by the authors.

[**Reply**] Thanks for the helpful and insightful comments regarding our manuscript. We appreciated that the reviewer recognized the unique value of our paper which may add to the growing knowledge regarding N impacts on forest ecosystems, especially in large areas of subtropical forests which are potentially making increasing contribution to carbon storage in China. The reviewer points out two limitations in this study. Firstly, the duration of the fertilization experiment was not accurately described. Data collected from March 2011 to July 2014 (and plants experienced 4 continuous growth seasons) were used in our study, so we briefly described the time scale of N fertilization to be 4 years in abstract. We accept the reviewer's suggestion and rephrased the duration of N fertilization to be 3.4

years in the manuscript.

Secondly, the reviewer pointed out that there were only three replications in each treatment. In fact, that the number of replications in our experiment was only three blocks was because of the actual distribution and topography of the subtropical forests. In eastern China, the distributions of subtropical forest stands are quite topografically fragmented, while relative flat stands are required to avoid N losses and minimize spatial heterogeneity among experimental treatments. Hence, after taking all the environmental conditions into consideration and comparing several evergreen broadleaved forests in subtropical regions, we determined to conduct N fertilization experiment in this forest located in the natural conservation zone of Guniujiang in Anhui Province, eastern China, because both the plant community and the landscape are good representatives of typical subtropical evergreen broadleaved forests. Actually, many of N addition experiments across different sites at boreal, temperate, tropical and subtropical forests have had similar number of replications (Rainey et al., 1999; Magill et al., 2004; Lu et al., 2010). For example, a similar experiment in a subtropical forest at Mt. Dinghushan in south China has a smaller plot size of 20 m×10 m and 3 replications (Lu et al., 2010). In the Hardward Forest where long-term N fertilization experiments have been conducted for more than 30 years, three replications of three N treatments (control: 0 kg N ha-1 yr-1, low N: 50 kg N ha-1 yr-1, high N: 100 kg N ha-1 yr-1) were settled. That is to say, our experimental treatments (e.g., design of N dosages and replications) are consistent or comparable with those in other regions of forests, which provided a good opportunity to compare results among sites and forest ecosystems globally. Moreover, the experiment introduced in our paper here is an important part of the Network of Nutrient Enrichment Experiments in China's Forest including 8 forests every ten degrees along latitude gradients in eastern China. We have conducted N fertilization experiment to stimulate N deposition simultaneously in 8 forests since 2011.

**[Specific comments]**
**[Comments]** 1. Environmental relevance - The application rates of 50 and 100 kg/N/ha are very high and I suspect are found in a few locations in China, but not likely widespread. My experience is that such high dosages almost always produce some effect but 1 year of 50 kg/N/ha is not the same as 5 years at 10 kg/N/ha. The authors should read a very good paper by Lovett and Goodale (2011) Ecosystems - that discusses this issue. Further, if my math is correct the authors are applying 100 kg N in 12 dosages per year, each time in 15 L of water. This makes 8kg N per time - dissolved in 15L, which is about 440g/L. Given the reportedly greater impacts of the treatments on the ground species, I am wondering about the direct effects of this spray? This should be discussed/evaluated.
**[Reply]** We agree with the reviewer's point that application rates of 50 and 100 kg/N/ha are high and found in a few locations in China. However, with the rapid growth of global population, Nr creation by human beings has increased approximately three times during 1850-2010 (Galloway et al.,

2014), of which large amount of reactive N emission lead to serious atmospheric N deposition, especially in eastern North America, Europe, China, India and Brazil (BassiriRad, 2015). In large parts of the non-urban areas across China, the rates of wet N deposition have exceeded 15 kg N ha$^{-1}$ yr$^{-1}$ from 1995 to 2007 (Du et al., 2014). Taking the increasing rates of N deposition in eastern China into consideration, we set the dosages of N fertilization to simulate the potential effects of high N deposition. Moreover, the design of N50 and N100 were kept in accordance with previous studies conducted in boreal forests, temperate forests across Europe and America, tropical forests and subtropical forests (e.g., Rainey et al., 1999; Högberg et al., 2006; Lu et al., 2010; Alvarez-Clare et al., 2013). The consistency of N fertilization provided a good opportunity to compare results among sites.

Regarding the concentration of dosages, total $NH_4NO_3$ was divided into 12 dosages and applied to the forest in each month at regular intervals during a year. According the design of N treatments (N50: 50 kg N ha$^{-1}$ yr$^{-1}$ and N100: 100 kg N ha$^{-1}$ yr$^{-1}$) and the size of plots (20 m$\times$20 m), $NH_4NO_3$ in dosages of 0.48 kg plot$^{-1}$ month$^{-1}$ and 0.95 kg plot$^{-1}$ month$^{-1}$ were dissolved in 15 L of fresh water, respectively, and then sprayed uniformly in N50 and N100 plots using a back-hatch sprayer. The unfertilized plots were similarly treated with 15 L of fresh water without $NH_4NO_3$. Therefore, the 0.48 kg and 0.95 kg $NH_4NO_3$ dissolved in 15 L of fresh water, respectively, represent N concentration of 11.1 g/L and 22.2 g/L in N50 and N100 plots, much lower than high concentration of 440 g/L as the reviewer calculated. For detailed calculation in a case of N100 plots, please see the following:

N concentration (g N L$^{-1}$ plot$^{-1}$ month$^{-1}$) for N100 plots (100 kg N ha$^{-1}$ yr$^{-1}$)

= 285.71 kg $NH_4NO_3$ ha$^{-1}$ yr$^{-1}$ (please note: 1 kg N = 2.8571 kg $NH_4NO_3$)

= 0.95 kg $NH_4NO_3$ plot$^{-1}$ month$^{-1}$

= 0.33 kg N plot$^{-1}$ month$^{-1}$

Therefore, the N concentration for each plot:

= 0.33 kg N /15L

= 22.2 g N L$^{-1}$ (please note: the amount of 0.33 kg N was dissolved into 15 L of fresh water for each plot and each month)

Therefore, to avoid misunderstanding, we described more details about the dosage of N fertilization to make it clear in the revised manuscript (Lines: 143-147).

[Comments] 2. This is a short study (3.4 years - should be consistent throughout which it isn't at present) with relatively low replication. In both instances real changes may be occurring but statistically they are not different among treatments. Throughout the paper the authors refer to differences among treatments - when in fact they are not significant (e.g. Figure 3). Over time or with more replication it could be true - just as equally it may still be noise in the system. The authors are guilty of talking about differences when in fact they statistically the same.

[Reply] We appreciate the reviewer's remind about the statistical result of the data. We described the limitation of the relatively short-term study (3.4 years) and the low replication (n=3) in our

experiment in "Materials and methods" of our revised manuscript. Regarding the replications settled in our experiment, the plots were limited by the actual area of the subtropical forests. As we reported above, the distributions of subtropical forests are quite fragmented, while relative flat forests are needed to avoid N losses and minimize spatial heterogeneity among plots. Hence, after comparing several forests in subtropical regions, we conducted N fertilization experiment here because both the plant community and the landscape are very good representatives of typical subtropical evergreen forests. Moreover, a similar experiment in another subtropical forest at Mt.Dinghushan in China has plot size of 20 m×10 m and replications of 3. Overall, the consistency in the design of N dosages and replications across boreal, temperate, tropical and subtropical forests including ours provided a good opportunity to compare results among sites and forest ecosystems globally. In addition, we carefully checked our description of the results, especially those regarding statistical analysis, and avoided misleading words in the revised manuscript. Please see the detailed revisions in Lines 227-258 at Page 7.

**[Comments]** 3. The main argument for the difference in response among growth forms is shading. There is no evidence for this presented in the manuscript (not measured). Equally, there is no evidence for statistical differences in N content among treatments (supplementary info). Thus while the authors present a mechanistic reason behind the differences there is no real statistical evidence to support these claims. Changes in canopy cover were not assessed and N or P (nothing else shown) are not significant among treatments. Soil pH is lower, but Al or Mn are not measured. I found the discussion section (4.3) very misleading for example - "total N content of soil was enhanced by N fertilization and P concentration in plant leaves and in fine roots showed that N concentration increased" - not only is this a poor sentence, it is factually incorrect -N content did not increase in the 50 Kg N treatment nor did N content significantly increase (Figures are actually labeled incorrectly). Similarly there is no evidence of P being lowered by the treatment (soil or plant). Why was nitrate or ammonium not measured?

**[Reply]** Many thanks for these comments. We checked and corrected the wrong labels in the figure. In the Discussion section, we made a substantial revision to discuss potential mechanisms underlying the different responses of different growth forms to N fertilization. First of all, to provide an evidence of shading or light availability, we added the data of canopy cover measured by a digital camera with a fisheye lens [lines: 202-205 at Page 6]. We used this results in the discussion as following: The results of estimated canopy cover showed no significant differences between unfertilized (0.77±0.01) and N fertilized plots (0.76±0.04 and 0.72±0.01 in N50 and N100 plots), suggesting that factors other than light availability had played a crucial role in understory plants in this old-aged subtropical forest during 3.4 years' N fertilization [lines: 329-336 at Page 9]. Secondly, we deleted the misleading sentences in section 4.3 and focused on the negative effects of potential N saturation on the growth of understory plants [lines: 338-355 at Page 9-10]. Actually, we have measured the changes of nitrate or

ammonium of 0-10 cm soil in N fertilized plots (please see the Figure S1). Because the concentrations of nitrate or ammonium were more easily influenced by temperature, moisture (precipitation) and spatial heterogeneity, we did not bring these data into analysis to support our results. Nevertheless, the general pattern of the responses of soil total N content to N fertilization was similar to soil mineral content.

Figure S1. Soil (0-10 cm) mineral nitrogen content (the sum of $NH_4$ and $NO_3$-N, mg/kg). (a) Seasonal variation of soil mineral nitrogen content (mean ± se) in unfertilized plots from May 2011 to May 2013, and (b) effects of nitrogen fertilization on soil mineral nitrogen content. Different labels in (b) indicate significant differences among three N treatments in the same month ($p<0.05$).

[Figure]

[Comments] 4. The P fertilizer study added at the end reads just like an add on and does not help the paper and it should be deleted. Similarly the text on lines 243-249 could be deleted. [Reply] Thanks. We added results from the P fertilizer study in the manuscript to provide data for the P limitation hypothesis in the subtropical forest. In the revised manuscript, we followed the reviewer's suggestion and deleted the initial Fig. 6 and the text on lines 243-249.

In addition, we mentioned the positive responses of plants to P fertilization in tropical and subtropical forests and included data from this P fertilizer study as a supplementary support [lines: 287-297]: As a supplement, we used data from a P fertilization experiment conducted in another subtropical forest with similar community structure nearby our experiment site to check if P limits plant growth. We applied 50 kg ha$^{-1}$ yr$^{-1}$ P ($P_2O_5$) to the forest and measured the growth of the dominant tree species (*C. sclerophylla*) following the same steps presented in the 'Materials and methods' section in this paper. After two years' P fertilization, we found that the annual absolute basal area increment and relative basal area in P fertilized plots were 56.0% and 101.5% higher, respectively, than in unfertilized plots (p=0.02 and p=0.03, respectively, unpublished data). Our results from N fertilization and the supplementary P fertilization experiments indicate that plant growth in subtropical forests might be highly limited by P, but this is in great need for further verification in the future studies.

**[Comments]** 5. The data shown in Figure 2 - basal area changes over time by size class are self-evident and this could be deleted. I am much more interested in how size class distribution compared among the study plots at the beginning of the study period. With such low replication (40 trees per plot = 120 trees per treatment, which then get broken down into smaller units - some of these comparisons may be being made on a very few trees). As addressing these comments should alter the paper substantially I will not comment on editorial issues.

**[Reply]** Many thanks for reviewer's suggestions. We deleted Figure 2 - basal area changes over time by size class in the revised manuscript as suggested.

**To Anonymous Referee #2:**

[**Major comments**]: My opinion is that the text in large parts of the paper needs to be rephrased. The results needs to be much more carefully described and the authors should make an effort in making it more clear what differences that are statistically supported and what are not. Several of the main results discussed (e.g. that N addition stimulated growth of large trees and suppressed growth by small ones) is not supported by data. I agree that it is likely that the suppression of understory vegetation stems from increased light competition with a denser overstory, and that this was caused by N addition, But this is NOT reflected by any of the data collected by the authors. Perhaps, N addition increased leaf area or canopy cover of the overstory, and by this suppressed light conditions and the growth of the understory? Such effect would over time be expected to be reflected by increased basal area but the limited duration of the experiment (3 and not 4-year as claimed in the text) may have been too short the capture such response. If there are any data on canopy cover or light transmission to the ground level, such data would definitely be worth exploring as it may help explaining the results. The addressed questions could easily be made a bit more sophisticated by asking for differences compared to the known response from other forest systems (e.g. temperate, tropical or boreal forests). This is partly related to how the available knowledge from other systems is described in the introduction (see comments further down). The last part of the abstract can be misleading as the result presented only supports that small trees grow better under ambient N than elevated N and there is actually no support at all for higher growth of large trees under elevated N. The last sentence of the abstract, i.e. the conclusion of the study/implication of the results is extremely vague as the reader is not provided with any clue to why it is important to consider more parts of the vegetation than just the trees. A hint may be given by the results presented, i.e. that large trees responded differently from other parts of the vegetation, but the authors never help the reader describing why this is problematic of what can happen if the response is just evaluated based on the trees (large

trees).    I miss information on whether the growth in study system in general is N limited. In my opinion this is essential information when it comes to evaluating the response to the N addition. If not, or if the growth in the system is co-limited by other nutrients, a lack of N response should be interpreted a bit differently than if N is the solely or main limiting nutrient. I believe that this is important as the response to the N treatment in general was rather weak and most often non-significant. In fact the additional data presented on P addition (Fig. 6) might suggest that P is co-limiting nutrient.

**[Reply]** Thanks very much for the constructive suggestions. We have made a substantial revision according to the reviewer's suggestions. First of all, we accept the constructive suggestion that whether the growth in this study system in general is N limited, which is the most important question to answer. Indeed, previous results from boreal and temperate forests have showed that most trees have a positive growth response and therefore higher potential C storage to N fertilization because the status of N limitation was largely alleviated by the increasing N inputs (e.g., Thomas et al., 2010; BassiriRad et al., 2015) [lines: 60-65]. On the contrary, in addition to the ubiquitous concept that P was a critical element driving plant growth in tropical forests (Vitousek et al., 1991), heterogeneous nutrient limitation concept that the growths of plants were co-limited by multiple nutrients has been proposed recently to explain why diverse plants respond differently to nutrient addition (Wright et al., 2011; Alvarez-Clare et al., 2013; Wurzburger & Wright 2015) [lines: 78-84]. Therefore, the patterns of specific nutrient limitation and responses of plants to added nutrients among diverse forest ecosystems need further exploration, especially in subtropical forests which were rarely investigated.

Secondly, according to our main focus on answering the question "whether N is limited in this old-aged evergreen subtropical forest" in the revised manuscript, we rewrote the Introduction and Discussion sections with a simple hypothesis: if the subtropical forest is limited by N, a positive response of trees ascribed to enhanced N fertilization but a negative response of understory growth forms to N fertilization due to the potential expansion of canopy crown and limitation of light availability. In the Discussion section, we have added an evidence of canopy cover as following: Further, Our results of forest canopy cover estimated by photographic fisheye showed no significant differences between unfertilized (0.77±0.01) and N fertilized plots (0.76±0.04 and 0.72±0.01 in N50 and N100 plots, respectively), which was consistent with the findings of Lu et al. (2010). Although the understory light irradiance fluctuated largely during a day and was very hard to detect precisely, our measurements of forest canopy cover provided a rough evaluation for light availability. The results might indicate that other factors in addition to the low light availability in this old-aged forest had also played a crucial role in influencing understory plants during 3.4 years' N fertilization. Moreover, We discussed the potential mechanisms underlying the contrasting responses of different plant growth forms to N fertilization, including potential P but not N limitation or heterogeneous nutrient limitation on trees in this subtropical forest as in tropical forests, low light availability for understory plants, and potential N saturation after 3.4 years' N fertilization [lines: 262-335].

**[Comments]** L. 43-53. The authors seriously exaggerates the lack of knowledge, and I would go so far as saying that the content of this paragraph gives a false picture the available literature on N effects in forested systems. First, studies from boreal areas are not at all limited to tree response. In fact there has been much other work done, both on other plant groups and on other organisms than plants. For a quick overview see the summary paper by Bobbink et al 2010 (that is cited elsewhere in the ms). Second, the authors claim that the response of forest understory communities rarely have been studied, which is simply not true. Just a few examples are van Dobben et al. 1999 (For Ecol Manag 114, 83–95); Strengbom et al 2001 (Funct Ecol 15, 451–457); Gilliam 2006 (Journal of Ecology 94: 1176–1191), and there are many more.

**[Reply]** Many thanks for reviewer's suggestions. We carefully reviewed available literatures about the effects of N fertilization (or deposition) on plants in boreal, temperate, tropical and subtropical forests. Then, we regrouped the introduction. We recognized many valuable studies conducted in boreal areas focusing not only on trees, but also on other plant growth forms, for example dwarf shrubs, herbaceous species and seedlings. We have synthesized more related literatures in the revised introduction. Please see lines 58-96 at page 3.

**[Comments]** L. 110-118. I can understand why you exclude trees that died, and understand why trees that had decreasing DBH were excluded (but not necessarily agree that they should be excluded, as you then only accept measuring errors in one direction but not the other), but how can you justify excluding trees that showed no change in DBH? I am very worried that by omitting trees that showed no change in DBH may have seriously have influenced the results of your study and risk exaggerating the positive response that the N addition may have had. The authors should in general be much more careful when presenting non-significant differences. If these at all should be mentioned it should be absolutely clear to the reader that these are non-significant differences. Much of the discussion, and even parts of the major conclusions, deals with non-significant differences that are presented as if they were statistically supported (e.g. L. 204-205, 210-212, 252-256).

**[Reply]** We checked all our data after reading the reviewer's comments. Definitely, our exclusion of trees that were dead, broken, had shrunk or did not have DBH changes, had a risk of exaggerating the positive response of trees to N fertilization. However, we found no significant difference between N treatments after including all the trees which were excluded at first and the addition of those trees did not change our results. It is likely that most trees that died, were broken, had shrunk or did not have DBH changes were small trees (DBH<5 cm) which earn a relatively small percentage of the total basal area and aboveground biomass. Nevertheless, to better and precisely report the results, we have re-analyzed the data (mainly the saplings, Figure 3) and described our results carefully, especially those showed no significant differences among N treatments.

**Fig. 3** Effects of N fertilization on the growth of saplings (mean ± se, n=3). (a) Absolute basal area increase, and (b) relative basal area growth rate. Numbers in these figures indicate the results of ANOVA.

[Figure]

**[Specific comments]**

**[Comments]** L. 22-23. There was no response at all of the larger trees! Avoid bringing up differences that are far from significant in the abstract. This is not just wrong, it is misleading! There is no description on how, when and why P was added in some plots.

**[Reply]** We appreciate the reviewer's comment. Our initial description focused much on the average values of basal area increment and RGR. In the revised manuscript, we revised the report of our result in abstract as following: On plot level, the absolute and relative growth rates of trees were not affected by N fertilization. Across the individuals of *Castanopsis eyrei,* the basal area increment of small trees with DBH (diameter at breast height) values of 5-10 cm declined by 50% in N fertilized plots, while the growth of median and large trees with DBH>10 cm showed no response to N fertilization. The growth rate of small trees and saplings and the biomass of understory shrubs and ground-cover ferns decreased significantly in N fertilized plots [lines: 32-38].

The description on how and why P was added in P-fertilized plots was described on lines 288-293 at page 8 as following: As a supplement, we used data from a P fertilization experiment conducted in another subtropical forest with similar community structure nearby our experiment site to check if P limits plant growth. We applied 50 kg ha$^{-1}$ yr$^{-1}$ P (P$_2$O$_5$) to the forest and measured the growth of the dominant tree species (*C. sclerophylla*) following the same steps presented in the 'Materials and methods' section in this paper. .

**[Comments]** L. 139-140. Do you have pre-treatment measures supporting that the vegetation was homogenous among plots at the initiation of the experiment? If so present these in a simple form. If not you should describe how the homogeneity was assessed.

**[Reply]** Thanks for the comments. We had a pre-treatment measure in March 2011 and evaluated the aboveground biomass of understory plants among the three N treatments. We presented these results in

the revised manuscript in [Lines 197- 200] at page 6 as following: Because the average aboveground biomass of shrubs/seedlings and ferns showed no significant differences across the three N treatments, we regarded the distribution of these understory shrubs/seedlings and ferns to be homogeneous among the three treatments before N fertilization in March 2011.

**[Comments]**: Table 1. It is not clear what the data represents. Are the numbers presented grand mean across all treatments? If so this should be clearly stated in the text explaining the table.

**[Reply]** We appreciate the reviewer's careful check. The data in Table 1 showed baseline data for four plant growth forms in this study before N fertilization. Numbers in the tables represent grand means (or mean ± standard error, n=9) of plants across all nine plots. We clearly stated these in the revised manuscript.

**[Comments]**: L. 157-158. I do not understand the results described here "The basal area and RGR of trees at the community level showed no significant response to N fertilization (Fig.1); however, the increase rates of basal area were likely hindered by N fertilization (Fig.1c)"What does this mean? As far as I can see from the statistical results presented and the data presented in Fig 1 there is just simply a lack of N response. Very unclear what you mean when saying that growth was hindered by N addition?

**[Reply]** We appreciate the reviewer's comments and sorry for the unclear description. We checked our description of the results, especially those with little significance through statistical analysis, and avoided misleading words in the revised manuscript.

In detail, we rephrased the text on lines 230-232 as following: Compared with the unfertilized plots, N50 and N100 fertilized plots showed a tendency toward higher averaged proportions of dead trees' aboveground biomass despite no statistically significant differences between them (Fig. 1d).

We rewrote the text on lines 240-243 as following: However, inconsistent with such negative responses of small trees to N fertilization, the basal area increment and RGR of median (DBH of 10-30 cm; see Fig. 2b-2c) and large trees (DBH >30cm; see Fig. 2e-2f) did not show significant responses to N fertilization ($p > 0.05$ in all cases).

**[Comments]**: L. 161-163. Be more careful when presenting non-significant differences. There might be a tendency towards more dead biomass under N addition but the difference is far from significant.

**[Reply]** We appreciate the reviewer's suggestions. We corrected the description of figure 1(d) as following: "Compared with the unfertilized plots, N 50 and N 100 fertilized plots showed a tendency toward higher averaged proportions of dead trees' aboveground biomass despite no statistically significant differences between them (Fig. 1d)" [Lines 230-232].

**[Comments]**: L. 164-165. The text here is wrong here. This result has nothing to do with the N treatment. The test and the fig just describes that basal area and RGR differed depending on size among individual trees of this species. This is very important as it seems like part of the conclusion is based on that there is a N effect here.

**[Reply]** Many thanks to the reviewers' comment on this figure. Initially, we aimed at reporting the result that basal area and RGR differed among individual trees with contrasting plant size. Small trees showed higher growth rate while larger trees showed lower growth rate. Then, the figure following this figure indicated different responses of the growth rate of trees in different sizes. In the revised manuscript, we deleted this figure to avoid the ambiguous description.

**[Comments]**: L. 168-173. The text here is in most parts misleading. The only effects that are sup-ported by the data presented is that the smallest trees growing under no N addition had higher basal area and higher RGR than small trees growing under N addition. All other differences that may or may not be visible in the figure is far from statistically supported and should not be mentioned here in the results.

**[Reply]** Many thanks for the reviewer's comments. We checked the description and we rewrote this part in the section 3.1 in Lines 232-241 at page 7 as following: Individuals of the dominant species *C. eyrei* with different initial DBH showed divergent responses of absolute basal area increments and RGR to N fertilization (Fig. 2a-2f). The small trees with DBH of 5-10 cm growing under unfertilized plots showed higher basal area increments and RGR than those small trees growing under N fertilized plots (Fig. 2a and 2d, $p<0.05$ and $p=0.03$, respectively). Specifically, the N50 and N100 fertilization decreased the absolute basal area increments of small trees at rates of 0.02 $m^{-2} ha^{-1} year^{-1}$ and 0.39 $m^{-2} ha^{-1} year^{-1}$, respectively. However, inconsistent with the negative responses of small trees to N fertilization, the basal area increment and RGR of median *C. eyrei* individuals with DBH of 10-30 cm and large *C. eyrei* individuals with DBH of >30cm showed no significant responses to N fertilization (Fig. 2b-2c and 2e-2f, $p>0.05$ in all cases).

**[Comments]**: L. 175-179. Is the test result presented in the fig correct? According the test results N addition influences RGR and mortality, but form the post hoc test there seem to be difference among the groups. From inspecting the data presented in the fig I wonder if there is some error among the letters indicating the differences among the groups in panel c and d. Results covering the data presented in fig 6 is missing from the result section

**[Reply]** Many thanks for the reviewer's comments. Similar to reply before, we have re-analyzed the data of the saplings. The results from *post hoc* test showed that although the annual absolute increments of basal area increments of saplings showed no significant response to N fertilization (Fig. 3a, $p=0.72$), the RGR of sapling growing in N50 and N100 plots relative to the unfertilized plots showed a substantial decrease at rates of 0.021 $m^{-2} m^{-2} yr^{-1}$ and 0.019 $m^{-2} m^{-2} yr^{-1}$, respectively (Fig. 3b, $p<0.001$) [Lines: 248-251 at page 7].

**[Comments]**: L. 192-194. What is the rationale for expecting a common positive response for all types of plants? To me this seems a bit naïve, given that forest plant communities often are size structured communities (see e.g. papers by Peter Grubb), and understory species than can be expected to be light rather than nutrient limited.

**[Reply]** Thanks for the reviewer's insightful comments. In the revised manuscript, we changed our hypothesis as following: We attempt to explore whether N is a limiting element in the old-aged evergreen subtropical forest. We hypothesize a positive response of trees to N fertilization, but a negative response of understory growth forms to N fertilization due to the expansion of canopy crown and consequent reduction of light availability [Lines: 101-104 at page 4].

**[Comments]**: L. 204-205. The first part of the sentence (large trees) is NOT supported by the results.

**[Reply]** Thanks for the reviewer's comment. We rewrote relevant parts in abstract, result and discussion in the revised manuscript to avoid unclear description. The results of large trees were rephrased in abstract as following: However, inconsistent with the negative responses of small trees to N fertilization, the basal area increment and RGR of median C. eyrei individuals with DBH of 10-30 cm and large C. eyrei individuals with DBH of >30cm showed no significant responses to N fertilization (Fig. 2b-2c and 2e-2f, p>0.05 in all cases).

**[Minor technical and language errors]**
The text is in need of some language edition. I just provide a few examples were the text need some re-phrasing. I have not paid that much attention to text editing as I believe that the paper need to be substantially revised before the paper can reach an acceptable standard.

**[Comments]**: L. 23-24: Small trees, saplings and particularly understory shrubs and ground-cover ferns suppressed seriously by increasing N fertilization: : : How are the suppressed? I am not very fond of the wording seriously as it is not a neutral wording. Better describe how large the difference was. L 21-24: the small trees with DBH (diameter at breast height) values of 5-10 cm were hindered by N fertilization: : :In what way was the small trees hindered?

**[Reply]** Thanks for the reviewer's comment. According to the reviewer's suggestions, we revised the text on lines 33-38 as following: "Across the individuals of *C. eyrei*, the basal area increment of small trees with a DBH (diameter at breast height) of 5-10 cm has declined by 50% in N fertilized plots, while the growth of median and large trees with a DBH of >10 cm has not significantly changed with the N fertilization. The growth rate of small trees and saplings and the biomass of understory shrubs and groundcover ferns decreased significantly in the N fertilized plots."

**[Comments]**: L. 24-24. : : :Proportion of mortality? Here it is better to write either the

mortality of plants were: : : or The proportion of plants that died:

**[Reply]** We appreciate the reviewer's good suggestion. We have changed the description of mortality throughout the whole manuscript as "the proportion of died plants".

**[Comments]**: L 177. Avoid evaluating your results in the result section by using wording such as severely here. Save that type of wording for the discussion. L. 180-185. There is no need to present mean values in text if these are shown in the figure 5. Do not present data twice, choose either to present then in text or in the fig.

**[Reply]** We appreciate the reviewer's good suggestion. In the revised manuscript, the description of results had been remarkably changed. The mean values presented in text have been deleted.

**References**

Alvarez-Clare, S., Mack, M.C., and Brooks, M.: A direct test of nitrogen and phosphorus limitation to net primary productivity in a lowland tropical wet forest, Ecology, 94, 1540-1551, 2013.

BassiriRad, H.: Consequences of atmospheric nitrogen deposition in terrestrial ecosystems: old questions, new perspectives, Oecologia, 177, 1-11, 2015.

Du, E.Z., and Liu X.J. in Nitrogen deposition, critical loads and biodiversity, eds. Sutton et al. Springer Science+Business Media Dordrecht, pp.49-56, 2014.

Galloway, J. N., Winiwarter. W., Leip, A., Leach, A. M., Bleeker, A., andErisman. J. W.: Nitrogen footprints: past, present and future, Environ. Res. Lett., 9, 115003, 2014.

Högberg, P., Fan, H.B., Quist, M., Binkley, D., and Tamm, C. O.: Tree growth and soil acidification in response to 30 years of experimental nitrogen loading on boreal forest, Global Change Biol., 12: 489-499, 2006.

Lu, X.K., Mo, J.M., Gilliam, F.S., Zhou, G.Y., and Fang, Y.T.: Effects of experimental nitrogen additions on plant diversity in an old‐growth tropical forest, Global Change Biol., 16, 2688-2700, 2010.

Magill, A. H., Aber, J. D., Currie, W. S., Nadelhoffer, K. J., Martin, M. E., McDowell, W. H., Melillo, J. M., and Steudler, P.: Ecosystem response to 15 years of chronic nitrogen additions at the Harvard Forest LTER, Massachusetts, USA., For. Ecol. Manage., 196, 7-28, 2004.

Rainey, S. M., Nadelhoffer, K. J., Silver, W. L., and Downs, M. R.: Effects of chronic nitrogen additions on understory species in a red pine plantation, Ecol. Appl., 9, 949-957, 1999.

Thomas, R.Q., Canham, C.D., Weathers, K.C., and Goodale, C.L.: Increased tree carbon storage in response to nitrogen deposition in the US, Nature Geosci., 3, 13-17, 2010.

Vitousek, P. M., and Howarth, R. W.: Nitrogen limitation on land and in the sea: how can it occur? Biogeochemistry, 13: 87-115, 1991.

Wright, S.J., Yavitt, J.B., Wurzburger, N., Turner, B.L., Tanner, E.V., Sayer, E.J., Santiago, L.S., Kaspari, M., Hedin, L.O., and Harms, K.E.: Potassium, phosphorus, or nitrogen limit root allocation, tree growth, or litter production in a lowland tropical forest, Ecology, 92,1616-1625, 2011.

Wurzburger, N., and Wright, S. J.: Fine‐root responses to fertilization reveal multiple nutrient limitation in a lowland tropical forest, Ecology, 96: 2137-2146. 2015.

---

## Author Comment (AC3) · 7 Feb 2017

[revised manuscript text omitted]

---

## Author Comment (AC4) · 7 Feb 2017

Dear Dr. Zaehle,

Thank you so much for reviewing our manuscript entitled "*Contrasting growth responses among plant functional types to nitrogen fertilization in a subtropical forest in China*" (bg-2016-416). The referees have made a number of insightful comments and suggestions. We have carefully addressed these comments/suggestions and revised the MS accordingly.

The reviewers' major comments are summarized as the following three points: (1) the referee #1 questioned that the dosage of our N fertilization was too high; (2) referee #1 commented on that our experiments were only three replications in each treatment; and (3) referee #2 asked that our hypothesis should be more clear and the explanations in Discussion section should be biologically sound.

For the first point, we have made more clear statements in the 'Materials and methods'. We are sorry that our simple description may cause referee #1 a misunderstanding and incorrect calculation of the N dosage. In fact, in our experiment, 0.48 kg and 0.95 kg $NH_4NO_3$ were dissolved in 15 L of fresh water, respectively; namely, N concentration in N50 and N100 plots was 11.1 g/L and 22.2 g/L, respectively, but not 440 g/L in the N100 plots as the referee calculated. Taking the N100 treatment as an example, the equation for the calculation of the N concentration is as following:

N concentration [g N $L^{-1}$ $plot^{-1}$ $month^{-1}$)] for N100 plots (100 kg N $ha^{-1}$ $yr^{-1}$)
= 285.71 kg $NH_4NO_3$ $ha^{-1}$ $yr^{-1}$ (please note: 1 kg N = 2.8571 kg $NH_4NO_3$)
= 0.95 kg $NH_4NO_3$ $plot^{-1}$ $month^{-1}$
= 0.33 kg N $plot^{-1}$ $month^{-1}$
Therefore, the N concentration for each plot:
= 0.33 kg N /15L = 22.2 g N $L^{-1}$ (please note: 0.33 kg N was dissolved into 15 L of fresh water for each plot and each month).

For the second point, that the number of replications in our experiment was only three blocks was because of the actual distribution and topography of the subtropical forests. In eastern China, the distributions of subtropical forest stands are quite topografically fragmented, while relative flat stands are required to avoid N losses and minimize spatial heterogeneity among experimental treatments. Hence, after comparing several forests in subtropical regions, we conducted N fertilization experiment here because both the plant community and the landscape are good representatives of typical subtropical evergreen broadleaf forests. Actually, many of N addition experiments across different sites at boreal, temperate, tropical and subtropical forests have had similar number of replications (Rainey *et al., Ecol. Appl., 9, 949-957, 1999*; Magill *et al., For. Ecol. Manage., 196, 7-8, 2004*; Lu *et al., Global Change Biol., 16, 2688-2700, 2010*). For example, a similar experiment in a subtropical forest at Mt. Dinghushan in south China has a smaller plot size of 20 m×10 m and 3 replications

(Lu *et al., Global Change Biol., 16, 2688-2700, 2010*). In the Hardward Forest where long-term N fertilization experiments have been conducted for more than 30 years, three replications of three N treatments (control: 0 kg N ha$^{-1}$ yr$^{-1}$, low N: 50 kg N ha$^{-1}$ yr$^{-1}$, high N: 100 kg N ha$^{-1}$ yr$^{-1}$) were settled. That is to say, our experimental treatments (e.g., design of N dosages and replications) are consistent or comparable with those in other regions of forests, which provided a good opportunity to compare results among sites and forest ecosystems globally.

For the third point, we rephrased our hypothesis and focused on answering the question of "whether N is limited in this old-aged evergreen broadleaved subtropical forest" and rewrote the Introduction and Discussion section. We carefully stated the results of our experiment, and added an evidence of canopy cover and predicted the potential risk of N saturation in the subtropical forests in the Discussion section.

The detailed responses to each question and comment of the reviewers are attached with this cover letter. We hope that our revision is satisfactory to you and the reviewers. Please contact me if you have any concerns or further information is requested. Thank you!

I am looking forward to receiving your decision on our manuscript.

Yours sincerely,
Jingyun Fang
Professor
Department of Ecology, Peking University
Beijing 100871, China
Tel/Fax: +86-10-6276 6560
E-mail: jyfang@urban.pku.edu.cn

---

## Author Response (AR2)

Authors' response to editor' comments on the manuscript bg-2016-416 "*Contrasting growth responses among plant growth forms to nitrogen fertilization in a subtropical forest in China*" by Di Tian et al.

Di Tian, tiandi@pku.edu.cn

Jingyun Fang, jyfang@urban.pku.edu.cn

**To the editor:**

Dear Dr. Zaehle,

Enclosed please find our revised version of the manuscript (bg-2016-416) "*Contrasting growth responses among plant growth forms to nitrogen fertilization in a subtropical forest in China*" for possible publication in *Biogeosciences.*

Thank you very much for your helpful comments and suggestions which allowed us to improve the manuscript. We have carefully studied the comments and rephrased the introduction, results and discussion in the updated version. The point-by-point responses and the revised manuscript with changes marked are attached with this letter at the bottom.

We hope that our manuscript is now acceptable for *Biogeosciences* and look forward to hearing from you soon. Thank you!

On behalf of all the authors,
Di Tian (tiandi@pku.edu.cn)
Jingyun Fang (jyfang@urban.pku.edu.cn)

[**Comment**] 1. you added fish-eye observations, but they need to be mentioned as a separate section in the results (including a figure or a table), and not buried in the discussion.

[**Reply**] Many thanks for your helpful and insightful comment. We added a separate table (i.e. Table 2) in the Result and described the effect of nitrogen fertilization on the canopy cover (Lines 238 to 239 at Page 8) as follows: The indicator of forest canopy (i.e. [1-Fmv]) showed no significant differences between unfertilized and fertilized plots with 3.4 years of N fertilization (Table 2).

Table 2 The indicators of canopy cover (i.e. [1-Fmv]) of the three treatments in our experiments. $n$ indicates the number of replicates; SE indicates the standard error.

| Treatment | $n$ | Canopy cover | |
| --- | --- | --- | --- |
| | | mean | SE |
| CK | 3 | 0.77 | 0.01 |
| N50 | 3 | 0.76 | 0.04 |
| N100 | 3 | 0.72 | 0.01 |

[**Comment**] 2. your text reads as if there had been largely no or a negative response, but this does not reflect Figure 2 panel c which suggests that the growth rate of large trees has almost doubled. Maybe this is not statistically significant, because of the large variation, but this is too large a difference to be ignored. This is one of the points where the limitations of the study design need to be more carefully addressed, as the small plot size and low replication reduce the statistical power of the experiment. It is important to mention the limitation of the design not only in the Methods section, but also in case it does affect the interpretation of the Results.

[**Reply**] Thank you very much for your insightful comment and suggestion. Indeed, our results showed the plot-averaged absolute and relative growth rates of basal area and aboveground biomass of trees were not affected by N fertilization. However, across the individuals of *C. eyrei*, the small trees declined by 66.4% and 59.5%, respectively, in N50 (50 kg N ha$^{-1}$ yr$^{-1}$) and N100 fertilized plots (100 kg N ha$^{-1}$ yr$^{-1}$), while the growth of median and large trees with a DBH of >10 cm has not significantly changed with the N fertilization. Moreover, the growth rate of small trees, saplings and the aboveground biomass of understory shrubs and ground-cover ferns decreased significantly in the N fertilized plots, indicating contrast responses of plant growth forms to N fertilization.

As you pointed out, the Figure 2 panel c did suggest that the growth rate of large trees has almost doubled and the significance of N fertilization on the large trees might be hidden by the large variation or low replication. We rephrased the interpretation of Figure 2 panel c in Result 3.2 at page 9 as follows: "As opposed to the negative responses of small trees to N fertilization, the basal area increment and RGR of median (DBH of 10-30 cm) and large *C. eyrei* individuals (DBH of >30 cm) showed no significant response to N fertilization, but the averaged growth rate of large *C. eyrei* individuals in N50 plots almost doubled the value of the corresponding large individuals in unfertilized plots (Fig. 3b-3c and 3e-3f, *p*>0.05 in all cases)" [Lines 261-266].

In addition, we added the discussion on the influence of low replication and high spatial heterogeneity on plant growth, especially of the large *C. eyrei* individuals in Discussion 4.1 as follows: "Furthermore, our observation of large trees with DBH >30 cm showed that the averaged growth rate of large *C. eyrei* individuals in N50 plots almost doubled the value of the corresponding large individuals in unfertilized plots. Nevertheless, the results of ANOVA showed that the effect was not significant. As the number of large trees in the experiment was relatively less than the small trees, the low replication and high spatial site heterogeneity might have reduced the statistical power of N fertilization on the large trees. Thus, fertilization experiments with more homogeneous plots and more replicates are warranted to further strengthen these findings. Overall, given the negative and potential positive effects of N fertilization on small and large trees, it is of urgent necessity to conduct long-term monitoring of the trees which would provide alternatives for accurately evaluating the forest dynamics under the enhanced global N deposition" [Lines 329-340].

[**Comment**] 3. Figure S1 provides good information in addition on what is going on at the sites, and should not be buried in the SI. Please include this, and a paragraph outlining the implications of this data into the main manuscript.

[**Reply**] Thanks for your suggestions. Yes, following your suggestion, we moved Figure S1 to the main context as Figure 1. We also added Figure 1 panel c to show the changes of soil N:P ratio with the effect of N fertilization. Corresponding to these changes, we added the descriptions on detail method and statistical analysis used in Figure 1 in the Methods section. Moreover, we added a new paragraph in Results to state the changes of soil N and P contents and pH as follows: "3.4 years of N fertilization significantly increased the N content of 0-10 cm soil (*p*=0.03), especially in N100 plots (Fig. 1a), but showed no significant effect on soil P content (Fig. 1b, *p*>0.05), thus leading to a significant increase in soil N:P ratio (Fig. 1c, *p*=0.02). Additionally, the N fertilization also decreased mildly soil pH and aggravated soil acidification (Fig. 1d, *p*=0.05)" [Lines 239-243].

[**Comment**] 4. As reviewer#1 already commented, please remove the reference to the unpublished material of the P fertilisation experiment (e.g. L289-298), because without proper explanation of the experimenet design, results and their uncertainty, these data appear unsubstantiated.

**[Reply]** Thank you. We have removed the references and related P fertilization experiment in the revised manuscript.

[**Comment**] 5. As reviewer #1 already commented, the environmental relevance of the magnitude of the fertilisation is questionable. Please down-tone any reference to N deposition in the introduction (e.g. first paragraph), as your experiment is not designed to address the likely response of N deposition on these ecosystems.

**[Reply]** Thank you for the helpful suggestions. We have carefully checked the whole text and rephrased the interpretation of N deposition to avoid misleading wording. Especially, we rephrased several sentences in Introduction section as well in Abstract, to down-tone the effects of N deposition.

[revised manuscript text omitted]

---

## Author Response (AR3)

Authors' response to editor' comments on the manuscript bg-2016-416 "*Contrasting growth responses among plant growth forms to nitrogen fertilization in a subtropical forest in China*" by Di Tian et al.

Di Tian, tiandi@pku.edu.cn
Jingyun Fang, jyfang@urban.pku.edu.cn

**To the editor:**

Dear Dr. Zaehle,

Enclosed please find our revised version of the manuscript (bg-2016-416) "*Contrasting growth responses among plant growth forms to nitrogen fertilization in a subtropical forest in China*" for possible publication in *Biogeosciences.*

We appreciate very much your insightful comments which allowed us to improve our manuscript. We have carefully studied your helpful suggestions and rephrased the title, results and discussion in the updated version. Following your comments, we have rephrased the title as *"Growth responses of trees and understory plants to nitrogen fertilization in a subtropical forest in China"* in the revised manuscript. The point-to-point responses and the revised manuscript with changes marked are attached with this letter at the bottom.

We hope that our manuscript is now satisfied with you and look forward to hearing from you soon. Thank you!

On behalf of all the authors,
Di Tian (tiandi@pku.edu.cn)
Jingyun Fang (jyfang@urban.pku.edu.cn)

**[Comment]** 1. The title focusses the comparison on growth form, whereas in reality, your results mainly give evidence for the differential response between over and understorey, with the latter including different growth forms. This needs to be adequately reflected in title and abstract.

**[Reply]** Many thanks for your helpful and insightful comment. Following you suggestions, we have rephrased the title as "Growth responses of trees and understory plants to nitrogen fertilization in a subtropical forest in China" in the revised manuscript. Then, we checked the wording in the abstract to adequately reflect different responses of trees and understory plants to nitrogen fertilization.

**[Comment]** 2. In the methods, you have removed the information on site N deposition, which is however, relevant. I would present this information next to precipitation and temperature estimates earlier in Section 2.1.

[Reply] Thanks. Yes, we presented the information on N depostion next to precipitation and temperature in the revision as follows: The amount of wet N deposition in this region was 5.9-7.3 kg N ha$^{-1}$ yr$^{-1}$ [Lines 121-122].

**[Comment]** 3. New Section 3.1: I think this section needs to be accompagnied by a statement as to whether it is possible that the fish eye measurements did not provide evidence for changes in total forest cover, while there still may be a shift between the contribution of overstorey and understorey trees to the total forest cover (partly therefore explaining the difference between the response across DBH?

[Reply] Thanks. We agree with you. Accordingly, we added these statements in the New Section 3.1 as follows: Although the fish eye measurements did not provide evidence for changes in total forest cover with the effects of N fertilization, there still may be a shift between the contribution of overstory and understory trees to the total forest cover [Lines 239-242]. In addition, we added a statement in the Discussion section as follows: our measurements of forest canopy cover provided a rough evaluation for light availability and a potential shift between the contribution of overstory and understory trees to the total forest cover which partly explained the difference between the responses across trees in different size (i.e. different DBH classes) [Lines 371-376].

**[Comment]** 4. Given that the pH at the end of the treatment is 4.1, whereas in Section 2.1 it was introduced as 4.6, this needs more explicit mentioning in Section 3.1 and the Discussion on other limiting factors. This is a change of 0.5 pH within 3.4 years, which in my view cannot be described as "mild".

**[Reply]** Many thanks for your insightful comment. We have removed the word "mildly" in line 247 in the revision. Due to the seasonal changes of soil pH in the subtropical forest, the value of 4.6 introduced in Section 2.1 represented the averaged value during a whole year of 2011 before the N addition. However, the values of soil pH reported in Section 3.1 showed the difference among three N treatments (i.e. control, N50 and N100) in July 2014 after 3.4 years' of N fertilization. As showed in Figure 1, the soil pH values in these three treatments were 4.35±0.04 (mean±se), 4.21±0.06, 4.09±0.06, respectively.

[revised manuscript text omitted]